# Polymer morphology and interfacial charge transfer dominate over energy-dependent scattering in organic-inorganic thermoelectrics

Pawan Kumar[1], Edmond W. Zaia[2,3], Erol Yildirim[4], D.V. Maheswar Repaka[1], Shuo-Wang Yang[4], Jeffrey J. Urban[2] & Kedar Hippalgaonkar[1]

Hybrid (organic-inorganic) materials have emerged as a promising class of thermoelectric materials, achieving power factors ($S^2\sigma$) exceeding those of either constituent. The mechanism of this enhancement is still under debate, and pinpointing the underlying physics has proven difficult. In this work, we combine transport measurements with theoretical simulations and first principles calculations on a prototypical PEDOT:PSS-Te(Cu$_x$) nanowire hybrid material system to understand the effect of templating and charge redistribution on the thermoelectric performance. Further, we apply the recently developed Kang-Snyder charge transport model to show that scattering of holes in the hybrid system, defined by the energy-dependent scattering parameter, remains the same as in the host polymer matrix; performance is instead dictated by polymer morphology manifested in an energy-independent transport coefficient. We build upon this language to explain thermoelectric behavior in a variety of PEDOT and P3HT based hybrids acting as a guide for future work in multiphase materials.

[1] Institute of Materials Research and Engineering, 2 Fusionopolis Way, Innovis, #08-03, Agency for Science, Technology and Research, Singapore, 138634 Singapore. [2] Molecular Foundry, Lawrence Berkeley National Laboratory, 1 Cyclotron Road, Berkeley, CA, 94720 USA. [3] Department of Chemical Engineering, University of California, Berkeley, 201 Gilman Hall, Berkeley, CA, 94720 USA. [4] Institute of High Performance Computing, 1 Fusionopolis Way, #16-16 Connexis, Agency for Science, Technology and Research, Singapore, 138632 Singapore. These authors contributed equally: Pawan Kumar, Edmond W. Zaia. Correspondence and requests for materials should be addressed to J.J.U. (email: jjurban@lbl.gov) or to K.H. (email: kedarh@imre.a-star.edu.sg)

New, emerging classes of organic semiconductors, polymers, and organic-inorganic composite materials have penetrated into areas of optoelectronics previously dominated by inorganic materials. Organic light emitting diodes have reached wide commercial availability, and organic-inorganic hybrid photovoltaics have shown an unparalleled rate of efficiency improvement[1,2]. Such solution-processable materials avoid the need for energy-intensive fabrication steps and instead utilize inexpensive, scalable, roll-to-roll techniques[3]. In particular, progress in this area has been driven by the development of materials based on poly(3,4-ethylenedioxythiophene) (PEDOT)[4,5]. PEDOT-based materials have demonstrated remarkably high conductivities, outperforming all other conductive polymer classes and driving tremendous interest in the use of soft materials in flexible electronics and thermoelectrics[6–8]. Soft thermoelectric materials can realize flexible energy generation or heating-cooling devices with conformal geometries, enabling a new portfolio of applications for thermoelectric technologies[3]. Of particular interest are hybrid soft nanomaterials – an emerging material class that combines organic and inorganic components to yield fundamentally new properties[9–12].

In hybrid systems, recent studies have focused on the use of strategies such as interfacial transport, structural/morphological effects, and modifications to the energy dependence of carrier scattering to improve electronic and thermoelectric performance[13–15]. However, it has proven difficult to establish the fundamental physics driving these performance enhancements. The most challenging of the proposed design strategies to evaluate is the role of energy dependent scattering, a phenomenon frequently and contentiously implicated in high performing thermoelectric materials[16,17]. More generally, advanced design of high performing soft thermoelectric materials has been stymied by the fact that transport in these systems is complex and resists description by a unified transport model. Development of next generation soft hybrid materials and modules requires improved understanding of the carrier transport physics in complex multiphase systems.

Kang and Snyder recently proposed a generalized charge transport model (henceforth referred to as Kang-Snyder Model) for conducting polymers, which marks a significant advance in the theoretical tools available to the soft thermoelectrics field[18,19]. In this model, the thermoelectric transport of conducting polymers has been modeled using energy independent parameter '$\sigma_{E0}$' and energy dependent parameter '$s$' over a large range of conductivity. Vital to the theory developed is the energy-dependent scattering parameter, $s$, which distinguishes the majority of conducting polymers ($s = 3$) from the class of PEDOT-based materials ($s = 1$). However, these two parameters are difficult to uniquely measure. Additionally, this first report did not cover hybrid organic-inorganic materials, which creates an important gap in the relevant material classes that requires further investigation.

Here, we apply this Kang-Snyder framework to a set of hybrid thermoelectric materials to identify the physics responsible for favorable thermoelectric transport in these systems. We begin with a model system of tellurium nanowires coated with PEDOT: PSS. This system was chosen due to its high thermoelectric performance (ZT ~ 0.1 at room temperature), facile solution-based synthesis, and well-defined single crystalline inorganic phase[10,11,13]. Further, we have previously shown that this material can be easily converted to tellurium-alloy heterowires[10]. For example, using copper as the alloying material results in PEDOT: PSS-Te-$Cu_{1.75}$Te heteronanowires. It has been observed that formation of small amounts of alloy subphases yields improvement in the thermoelectric performance of these materials. However, whether the observed transport properties are dictated by the organic phase, inorganic phase, or interfacial properties is an open question. Here, we use a full suite of experiments, transport modeling, molecular dynamics, and first principles calculations to describe the exact nature of the organic-inorganic interactions. We also show that the Kang-Snyder framework can be applied effectively to hybrid systems, extending the theoretical tools available to experimentalists. In this way, we seek to fill an important gap in knowledge in this body of literature and inform the direction of future experimental work.

## Results

**Synthesis and Structure**. Synthesis of tellurium nanowires coated in PEDOT:PSS (PEDOT:PSS-Te NWs) and conversion to Te-$Cu_{1.75}$Te heterostructures (PEDOT:PSS-Te($Cu_x$) NWs) were performed as previously reported (Details in Methods)[10,11]. The synthesis of the tellurium nanowires is performed in the presence of PEDOT:PSS, which has been posited to act both as a stabilizing ligand and as a structure-directing agent (Fig. 1a)[11,13]. Te-$Cu_{1.75}$Te heterowires have a curved appearance as a result of alloy domains appearing at 'kinked' portions of the wires. Representative high-resolution transmission electron micrographs (HR-TEM) of the Te and $Cu_{1.75}$Te domains (Fig. 1c, d) show the highly crystalline inorganic phases and the different local morphologies. Note that the additional copper loading increases the total number of 'kinked' portions (Fig. 1b), while each 'kinked' portion is stoichiometrically stable and close to the $Cu_{1.75}$Te phase.

**Morphology and Interactions**. To probe the organic-inorganic interactions involved in carrier transport in this material system, we perform extensive Molecular Dynamics (MD) simulations and Density Functional Theory (DFT) calculations. Specifically, MD simulations uncover detailed information about adhesion and polymer morphology/structural changes in the vicinity of the Te nanowire and $Cu_{1.75}$Te hetero-nanowire surfaces. These analyses strongly suggest that structural templating effects occur during synthesis. Templating effects have been widely hypothesized to occur in such processes, but direct evidence has been lacking so far[13]. As a complement to our structural analyses, DFT is used to investigate the electronic effects that arise at the organic-inorganic interface. In particular, we estimate the amount of charge transfer between the organic and inorganic phases and probe the evolution of the electronic Density of States (DOS).

**Molecular dynamics (MD) simulations of the organic-inorganic interface**. The MD simulations (details in Methods and Supplementary Note 1) reveal self-alignment of PEDOT chains at the organic-inorganic interface for both Te and $Cu_{1.75}$Te NW surfaces. This self-alignment is only observed for the PEDOT chains in the vicinity of Te and the $Cu_{1.75}$Te surfaces, while the PSS remains unaligned. This phenomenon is clearly distinguished by comparing the structures and concentration profiles before and after simulated annealing (Fig. 1e, f, Supplementary Movies 1-2, Supplementary Figure 1). Further annealing simulations on PEDOT:PSS, pristine PEDOT and pristine PSS suggest that only first few layers of PEDOT moieties tend to align in a planar orientation on the inorganic surfaces (detailed discussion in Supplementary Note 1, see also Supplementary Figures 1-2, Supplementary Movies 1–5). Te/$Cu_{1.75}$Te nanowires are coated with ~2 nm thin PEDOT layer, which also corresponds to only a few layers (Supplementary Figure 1). It is also determined that the self-assembly and percolation of PEDOT chains are reduced on the kinked $Cu_{1.75}$Te surface (Supplementary Figure 3, Supplementary Movies 6-7).

To investigate the driving force for PEDOT alignment (and lack thereof for PSS) on the inorganic surface, the interfacial PEDOT-inorganic and PSS-inorganic interaction energies were calculated and compared. Two cells were equilibrated with six π-stacked PEDOT and three PSS oligomers (details in Methods, Supplementary Figure 4). In one of the cells, PEDOT chains are placed at the organic-inorganic interface; in the other, PSS chains. The polymer-Te interaction energy is determined to be −423 kcal/mol for PEDOT layers on the Te NW surface and −192 kcal/mol for the PSS oligomer on the Te NW surface. This result indicates a thermodynamic driving force for self-assembly of PEDOT over PSS on the nanowire surface, consistent with the structures observed in the MD simulations described previously. The interaction of the same systems with a $Cu_{1.75}Te$ NW is about two times stronger calculated as −792 and −388 kcal/mol for PEDOT layers on the NW surface and PSS layers on the NW surface respectively.

Simulated annealing (details in Methods) reveals that PEDOT chains tend to align in a planar configuration on both Te and $Cu_{1.75}Te$ surfaces, although self-assembly is observed on the Te surface and not the $Cu_{1.75}Te$ (Supplementary Figure 5). We attribute this phenomenon to stronger interaction between PEDOT and the $Cu_{1.75}Te$ surface, resulting in reduced movement of the PEDOT chains on the $Cu_{1.75}Te$ surface (Supplementary Movies 8-9).

**Density functional theory (DFT) calculations to probe nature of interactions at the interface.** To complement our understanding of polymer templating on the inorganic NW surfaces, DFT was used to calculate adsorption energies of PEDOT/PSS on these surfaces. Here, we consider a charged polaronic PEDOT hexamer $(EDOT_6)^{+2}$ and deprotonated PSS oligomer $(SS_6)^{-2}$ in a planar configuration close to the inorganic surface, as is predicted from MD simulations (Supplementary Note 2). At the Te NW-organic interface, adsorption energies of −0.42 eV and −0.31 eV per monomer were obtained for charged PEDOT and PSS, respectively, consistent with MD results that the Te surface is primarily occupied by adsorbed, aligned PEDOT. On the other hand, the PEDOT-$Cu_{1.75}Te$ adsorption energy is −0.56 eV, indicating an even stronger interaction.

Since MD and DFT both demonstrate the importance of the PEDOT-inorganic interaction at the interface in this system, further calculations were performed to determine the density of states and charge density difference at this interface. Figure 2a depicts the calculated charge density difference between $(EDOT_6)^{+2}$ and the Te surface, where a decrease in the electron density is represented in blue and electron density enrichment in red. Furthermore, using these charge density differences, charge transfer rates were extracted, yielding important insight into the electronic effects occurring at the organic-inorganic interface.

The maximum charge transfer rates from the inorganic surface to the first layer of (neutral) PEDOT chains on the surface were determined to be −0.078 and −0.144 for Te and $Cu_{1.75}Te$, respectively. Charge transfer rates are higher for the charged $(EDOT_6)^{+2}$ bipolaron, calculated to be −0.186 and −0.239 for Te and $Cu_{1.75}Te$, respectively (Supplementary Table 1). Note that a negative quantity here represents electron transfer from the inorganic surface to the organic PEDOT chains (i.e. hole transfer from the organic PEDOT chain to the inorganic surface). In every case, this charge transfer provides a de-doping effect of holes in the p-type PEDOT chains, which plays a key role in

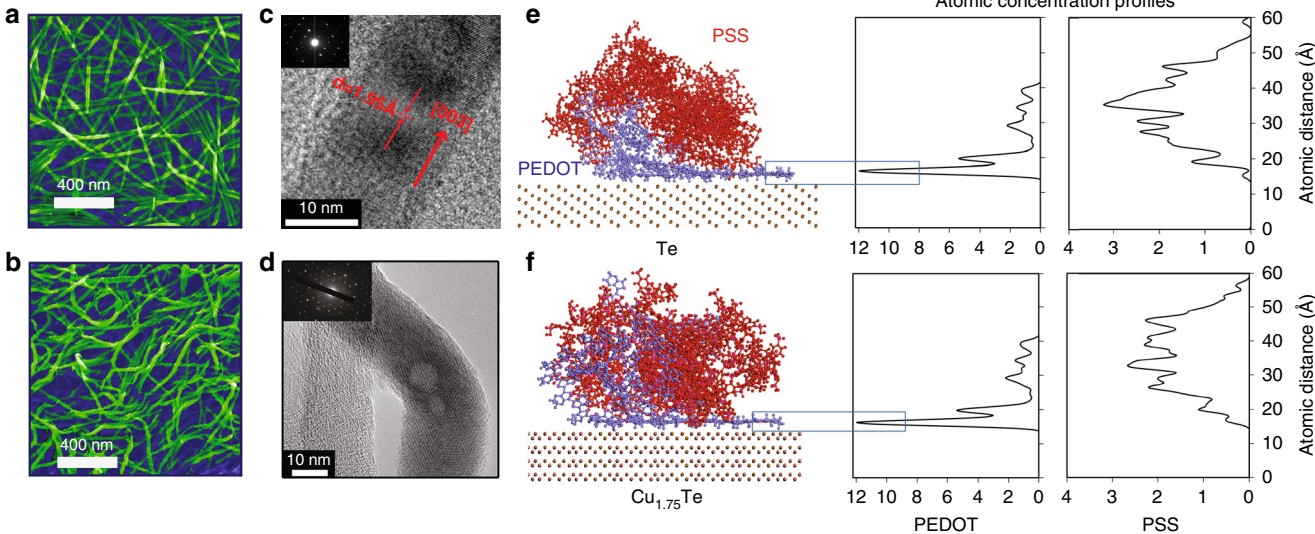

**Fig. 1** Morphology of hybrids and alignment of PEDOT:PSS at the inorganic interface (**a**, **b**) False-color scanning electron microscope (SEM) images of (**a**) PEDOT:PSS-Te and (**b**) PEDOT:PSS-$Cu_{1.75}Te$ films illustrate the overall morphology of the hybrid films – inorganic nanowires in a PEDOT:PSS matrix. The green color shows the surface nanowires and blue illustrates the 3D plane underneath, where the PEDOT:PSS polymer matrix is transparent (and hence invisible) in the SEM. (**c**, **d**) Representative high-resolution transmission electron microscopy (HR-TEM) images of (**c**) straight Te domains and (**d**) kinked $Cu_{1.75}Te$ alloy domains confirm the identity and crystallinity of these two phases. The insets show selected area electron diffraction (SAED) patterns consistent with the identified crystal structures. (**e**, **f**) MD simulations elucidate the polymer morphology and alignment at the organic-inorganic interface. Here, the final polymer structures are depicted after simulated annealing of five chains of $EDOT_{18}$ and $SS_{36}$ on (**e**) Te and (**f**) $Cu_{1.75}Te$ surfaces, both accompanied by respective atomic concentration profiles. The polymer concentration profiles are tracked using the atomic concentration of S in either PEDOT or PSS. There is a high concentration of S atoms in PEDOT observed at 3–5 Å from the nanowire surfaces, suggestive of highly ordered and aligned PEDOT chains at the organic-inorganic interface. Similar structures and concentration profiles were observed for both Te nanowires (NW) and $Cu_{1.75}Te$ heteronanowires (Supplementary Figures 1-2, Supplementary Movies 1-5), however, though alignment occurs, self-assembly of chains is reduced on the kinked $Cu_{1.75}Te$ surface, unlike on the Te surface (Supplementary Figure 3, Supplementary Movies 6-7)

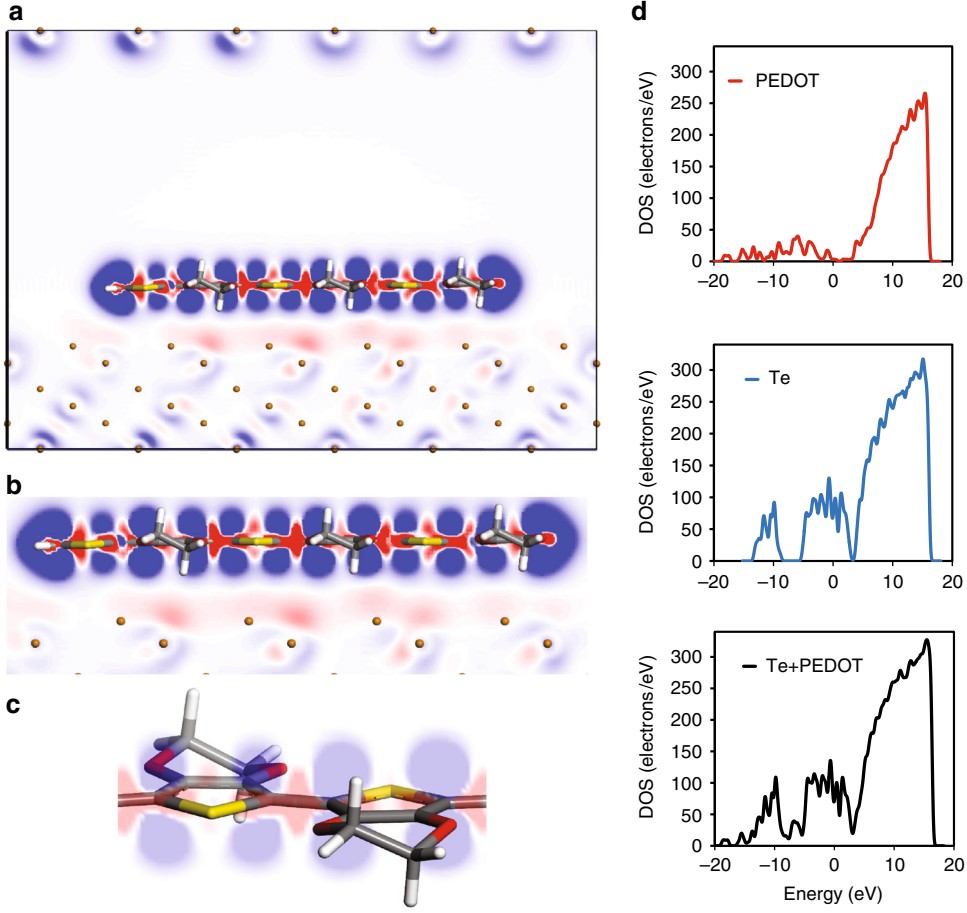

**Fig. 2** DFT calculations reveal electronic effects at the organic-inorganic interface. **a** Charge density redistribution within polaronic PEDOT hexamer $(EDOT_6)^{+2}$ on the Te surface, as calculated by the difference in total charge density with NW surface charge density and hexamer charge density as subsets (**b**) Electron transfer from Te surface to PEDOT chains monitored by increase of charge density (red) at the interface and decrease of charge density (blue) at the Te phase (**c**) Detailed visual illustrating an increase of charge density (red) within PEDOT monomer bonds and σ-orbitals of PEDOT carbon atoms and a concomitant decrease (blue) observed for the π-orbitals of PEDOT carbon atoms on Te NW surface (**d**) Density of States (DOS) calculated individually for PEDOT and Te compared with Te-PEDOT hybrid. The DOS of the hybrid is not renormalized due to minimal charge transfer across the interface

**Table 1 Interfacial charge transfer calculations**

| | De-doping effect[a] (electron/cm³) |
|---|---|
| Neutral PEDOT$_6$ on Te | $-6.19 \times 10^{20}$ for 3.6 Å |
| | $(-1.27 \times 10^{20}$ for 8 Å) |
| | (none > 15 Å) |
| PEDOT$_6^{+2}$ on Te | $-1.56 \times 10^{21}$ |
| Neutral PEDOT$_6$ on Cu$_{1.75}$Te | $-1.14 \times 10^{21}$ |
| PEDOT$_6^{+2}$ on Cu$_{1.75}$Te | $-2.05 \times 10^{21}$ |

De-doping level of neutral and doped PEDOT hexamer by Te and Cu$_{1.75}$Te surfaces for the optimized structures
[a]PEDOT monomer volume $1.26 \times 10^{-22}$ cm³

understanding the thermoelectric trends in these hybrid systems (Table 1). This de-doping effect is stronger for the doped PEDOT bipolaron compared to pristine PEDOT chains and also stronger for PEDOT on the Cu$_{1.75}$Te surface compared to PEDOT on the Te surface. The charge transfer and de-doping effect is only observed for the first two layer of PEDOT chains and vanished for higher distances.

As for the nature of the bonding between the organic and inorganic components, the weak charge density difference and atomic distances between organic and inorganic constituents, calculated to be between 3.6–4.0 Å, are analogous to other material systems that exhibit physical adsorption[20]. This conclusion is further corroborated by the Density of States (DOS) calculated for PEDOT, the Te surface, and the hybrid structure (Fig 2, Supplementary Figures 6-7 for PDOS), which depicts a trivial change in DOS distribution between the individual and hybrid structures. Interestingly, there is indeed an intra-chain charge density difference within the PEDOT chain at a field isovalue of 0.005 electron/Å³ (details in Methods). This result suggests electron density transfer from the PEDOT π orbitals to the σ orbitals, consistent with electron repulsion also known as a pillow effect[13,17,21,22], which occurs when an organic molecule approaches a metal surface (Fig 2a–c, Supplementary Figure 6b)[13,17,21,22]. Here, Pauli repulsion between the electron cloud of the nanowire surface and the PEDOT chain causes redistribution of electrons by repelling the electron cloud of the softer molecule (Supplementary Figure 8). As a result, the C-C bond distance between $(EDOT_6)^{+2}$ monomers is reduced from 1.42 to 1.40 Å when in proximity to the Te surface, characteristic of a benzoid to quinoid chain conformation change[23]. This conformational switch is also consistent with prior Terahertz spectroscopy studies that indicate charge-trapping near the polymer-Te nanowire surface[24,25].

Combined with the MD results, the DFT calculations strongly suggest that the interaction between the PEDOT and Te/Cu$_{1.75}$Te surface is a templating effect; charge transfer does occur at this interface, but no chemical bonding takes place between the organic and inorganic phases.

Hence, we expect that thermoelectric behavior in p-type PEDOT-Te hybrids is dominated by transport through the organic PEDOT matrix. This conclusion runs contrary to previous reports that have instead emphasized the role of the inorganic phase or *change in the* energy-dependent carrier scattering at interfaces as key drivers of the thermoelectric properties of hybrid materials[14,15,26]. Additionally, our results depict that the organic-inorganic interface in such hybrid systems is rich in aligned and extended PEDOT chains in addition to intra-chain charge redistribution. We conclude that alignment of PEDOT molecules at the organic-inorganic interface and charge transfer at the interface both play key roles in the high thermoelectric performance observed in the PEDOT:PSS-Te hybrid system, building upon earlier hypotheses proposing increased electrical conductivity at the interface[11].

**Seebeck and electrical conductivity analysis.** Thermoelectric properties and modeling using Kang-Snyder model: Armed with structural and morphological insight, we now correlate our measurements of thermoelectric transport with the observed and simulated structures. Figure 3a depicts the electrical conductivity and Seebeck coefficient of the PEDOT:PSS-CuTe hybrid system as a function of copper loading. In the absence of a robust

transport model for hybrid systems, researchers have previously had to rely on mean field theories. One commonly used model for multicomponent hybrids/composites is effective medium theory, which predicts that the Seebeck coefficient of the composite must lie between that of the individual materials (~190 µV/K for PEDOT:PSS-Te and ~10 µV/K for PEDOT:PSS-Cu$_{1.75}$Te at room temperature)[25]. Effective medium theory, therefore, fails to capture the observed enhancement of the Seebeck coefficient at low (~5%) Cu loading. This deviation had originally been speculated to be due to a change in the energy-dependence of carrier scattering upon introduction of Cu. Also, while the Cu loading in the composite is being varied, the overall inorganic content (Te and Cu) is controlled (typically 60–80%)[10,13].

The recently published Kang-Snyder model provides an opportunity to clarify this conundrum. Kang and Snyder showed that this framework handles pure polymeric systems well (including PEDOT); this makes the PEDOT:PSS-CuTe system a suitable candidate, since the PEDOT domains are known to be pivotal for charge transport in these hybrid materials. Further, PEDOT:PSS-CuTe is an excellent test case, since the effects of energy dependent scattering, (de-)doping, and morphology intermingle in a complex fashion. Given that the model independently treats the energy-dependent scattering (through the parameter *s*), doping (through the reduced chemical potential η), and energy-independent transport parameter $\sigma_{E_0}(T)$, such an analysis can provide insight that is both critical and previously inaccessible. According to this model, energy dependent conductivity, $\sigma_E(E, T)$ can be written as:

$$\sigma_E(E, T) = \sigma_{E_0}(T)\left(\frac{E - E_t}{k_B T}\right)^s \tag{1}$$

such that the total conductivity is given by:

$$\sigma = \int_0^\infty \sigma_E(E, T)\left(-\frac{\partial f}{\partial E}\right)dE \tag{2}$$

Using $\sigma_E$ from Eq. 1 and integration by parts, the total conductivity can be written as:

$$\sigma = \sigma_{E_0}(T) \times sF_{s-1}(\eta)$$

where $F$ is Fermi integral and $\eta = \frac{E_F - E_t}{k_B T}$ is reduced chemical potential and $E_t$ is the transport energy, below which there is no contribution to the conductivity even at finite temperature. The corresponding Seebeck coefficient can be written as:

$$S = \frac{k_B}{e}\left[\frac{(s+1)F_s(\eta)}{sF_{s-1}(\eta)} - \eta\right] \tag{3}$$

The η value was determined by using experimental Seebeck coefficient in Eq. 3 for a particular value of *s*. When applying the model for pure polymers, Kang and Snyder observed that traditional semiconducting polymers (e.g. polyacetalyene) follow $s = 3$ dependence, whereas PEDOT-based systems exhibit $s = 1$ dependence. In hybrid systems, it can therefore be presumed that, if transport in the polymer phase dominates the overall material properties, the energy dependence of transport (i.e. *s*) will remain the same as for the pure polymer matrix. If, on the other hand, transport in the hybrid material is modified by a change in the energy dependence of carrier scattering, it would be expected that *s* would also change. Therefore, to validate the hypothesis that the Seebeck enhancement observed in the PEDOT:PSS-Te(Cu$_x$) system is due to altered energy dependence of scattering, it is necessary that a change in the parameter *s* is observed upon introduction of Cu.

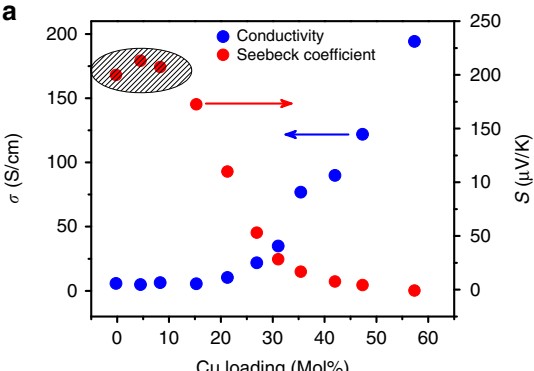

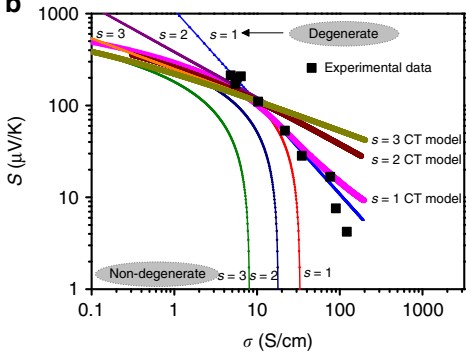

**Fig. 3** Kang-Snyder transport model applied to the PEDOT:PSS-Te(Cu$_x$) hybrid system. **a** Electrical conductivity and Seebeck coefficient at room temperature of the PEDOT:PSS-Te(Cu$_x$) NW hybrid system as a function of copper loading [Adapted with permission from Zaia et. al., **Nano Lett. 16**, 3352 (2016). Copyright (2016) American Chemical Society.][10] (**b**) The Kang-Snyder charge transport model with different energy-dependent scattering exponent, s, as described in the main text is shown. Our experimental data lie on the s = 1 curve, similar to pure PEDOT

**Table 2 Summary of physical phenomena contributing to thermoelectric trends**

|  | Templating effect | De-doping due to charge transfer at organic-inorganic interface (CT) | Cu loading | Total Conductivity and Seebeck coefficient | Discussion |
|---|---|---|---|---|---|
| PEDOT:PSS | X | X | X | X | X |
| PEDOT:PSS-Te | $\mu$ (↑) (strong) | $n_{de\text{-}doping}$ (↓) $S_{de\text{-}doping}$ (↑) (strong) | X | $\sigma(↑) = n_{de\text{-}doping}(↓) \times e \times \mu(↑)$ $S(↑) \propto \dfrac{1}{n_{de\text{-}doping}(↓)}$ | Conductivity increases due to templating; Seebeck increases due to charge transfer induced de-doping—higher than PEDOT:PSS |
| PEDOT:PSS-Cu$_{1.75}$Te (low Cu loading) | $\mu$ (↓) (weak) | $n_{de\text{-}doping}$ (↓) $S_{de\text{-}doping}$ (↑) (strong) | $n_{doping}$ (↑) $S_{doping}$ (↓) (weak) | $\sigma(↓) = \mu(↓) \times e \times \{n_{de\text{-}doping}(↓) + n_{doping}(↑)\}$ $S(↑) \propto \dfrac{1}{\{n_{de\text{-}doping}(↓) + n_{doping}(↑)\}}$ | Conductivity decreases due to further drop in templating; Seebeck increases due to stronger de-doping |
| PEDOT:PSS-Cu$_{1.75}$Te (High Cu loading) | $\mu$ (↓) (strong) | $n_{de\text{-}doping}$ (↓) $S_{de\text{-}doping}$ (↑) (weak) | $n_{doping}$ (↑) $S_{doping}$ (↓) (strong) | $\sigma(↓) = \mu(↓) \times e \times \{n_{de\text{-}doping}(↓) + n_{doping}(↑)\}$ $S(↓) \propto \dfrac{1}{\{n_{de\text{-}doping}(↓) + n_{doping}(↑)\}}$ | Conductivity increases due to strong Cu + induced doping; Seebeck decreases due to stronger Cu + induced doping |

Summary of the effects and interplay between templating, de-doping at the inorganic-organic interface and Copper loading, considering first PEDOT:PSS-Te and second PEDOT:PSS-Cu$_{1.75}$Te, and description of their respective roles on thermoelectric charge transport

For this goal, the Seebeck coefficient of the PEDOT:PSS-Te (Cu$_x$) hybrid system is plotted as a function of the conductivity (Fig 3b) and fit to the Kang-Snyder Charge Transport model[18]. We observe that the experimental data lie on the $s = 1$ CT model curve (Fig 3b) with $\sigma_{E_0}(T) = 5.47\ S/cm$. Thus, the $s$ dependence is unchanged between the pure PEDOT:PSS and its hybrid. Note, however, that while the Kang-Snyder model captures large trends in the S vs σ curve, small changes such as electron filtering cannot be isolated. Hence, in order to understand the non-monotonic trend in the Seebeck and conductivity, we study in detail the effect of (de-)doping and templating on the hybrid system (Table 2). Combining our experimental and theoretical results, we conclude that the complex thermoelectric trends of these hybrid films are dictated by the interaction of several effects. First, as suggested by extensive MD simulations, upon the formation of PEDOT:PSS-Te NWs, there is a templating effect for PEDOT moities on the inorganic surface. This phenomenon results in an increase of hole mobility in the interfacial polymer, increasing the electrical conductivity of the PEDOT:PSS-Te composite relative to the pristine polymer. This templating effect is weakened by the addition of Cu, which disrupts the inorganic surfaces and produces "kinked" inorganic morphologies. Secondly, detailed DFT calculations are indicative of charge transfer between the organic and inorganic phases, resulting in a de-doping effect of the $p$-type PEDOT chains (Table 1 and Supplementary Table 1). In the low Cu loading regime, this de-doping effect is relatively strong, and contributes directly to the increased Seebeck coefficient and moderately decreased electronic conductivity as observed here. This is contrary to previous hypotheses that a change in the energy dependence of carrier scattering is solely responsible for the non-monotonic thermoelectric trends observed in this range.

Upon further addition of Cu, a third effect emerges; Cu loading above 10% is associated with an increase in η, with only a nominal change in the $\sigma_{E_0}$ value (Table 3). This trend indicates that the addition of Cu introduces carriers into the film and modifies the transport through a doping channel. Previous reports on this material system have suggested that positively charged Cu ions, in addition to reacting with the inorganic phase, also remain in the PEDOT phase as ionic species. These remaining Cu ions likely interact with the PEDOT chains to increase the carrier concentration in the organic phase. This effect dominates at high Cu loading, which is associated with a strong increase in the reduced chemical potential. Note that $s = 2$ and $s = 3$ do not fit the experimental data for any value of the transport coefficient, $\sigma_{E_0}(T)$ (Fig 3b is plotted on log-log scale). While, for a fixed $\sigma_{E_0}$, η is modulated by charge redistribution between the organic and inorganic phases and doping from Cu

ions, only a change in morphology (templating, or kinked surfaces) can change $\sigma_{E_0}$.

**Validation of Kang-Snyder model for other PEDOT and P3HT based hybrid films**. In order to determine if this is generally true for PEDOT:PSS based films, we applied the Kang-Snyder model to different systems (Fig. 4a). Half-closed circles (olive) symbols show Seebeck and conductivity data on PEDOT:PSS films that are doped using an electrochemical transistor configuration[27]. Here, PEDOT:PSS is tuned by changing its oxidation state (de-doping) to obtain the optimal power factor, with presumably no change to the PEDOT morphology. Electrochemically doped PEDOS-C6 (a derivative of PEDOT) also exhibits Seebeck and conductivity data (purple close triangles) that lie on the $s=1$ curve with same $\sigma_{E_0}(T)$ value[28]. Open square (pink) symbols represent the Seebeck coefficient as a function of conductivity for PEDOT:Tosylate (Tos) system[4], where the insulating PSS polyanions are replaced by the small anion Tos, which improves inter-chain π-π interaction of PEDOT chains. The PEDOT:Tos Seebeck and conductivity data also lie on the s = 1 curve, albeit with a larger value of $\sigma_{E_0}(T)$. This is expected due to better alignment of PEDOT chains evidenced by Grazing Incidence Wide-Angle X-ray Scattering (GIWAXS) where the PEDOT:Tos contains well-ordered crystallite grains surrounded by some amorphous PEDOT:PSS regions. This is distinctly improved from the electrochemically doped PEDOT samples described above. $\sigma_{E_0}(T)$ value is further improved in the PEDOT:Tos-Pyridine + triblock copolymer system by controlling the oxidation rate as well as crystallization of oxidized PEDOT which further reduces film defects[29].

Interestingly, in comparison with pure organic derivatives of PEDOT, PEDOT-based hybrid films (half closed circles, closed stars and half closed shades in Fig. 4a)[10,26,30] also follow the $s = 1$ curve, with either the same value of $\sigma_{E_0}(T)$ as the pure polymer or higher (Table 3). For a second batch of PEDOT:PSS-CuTe samples, 0% Cu loading (closed red circle), the Seebeck and conductivity data lies on the same $s = 1$ curve as for Zaia et. al[10]. Here, we observe that as the Cu loading increases to 10%, the Seebeck is enhanced and the conductivity decreases (closed blue circle). This lower value of $\sigma_{E_0}(T)$ is consistent with curvature arising from 'kinked' Cu$_{1.75}$Te phase formation within the heteronanowires (Fig. 1d) as well as reduced movement of the PEDOT on the Cu$_{1.75}$Te (Supplementary Movie 9 and Supplementary Figure 7). The associated Seebeck increase is due to a de-doping effect, as η is observed to decrease slightly when Cu loading is increased from 0 to 10%. With further loading of Cu, the Seebeck decreases and the conductivity increases (further details about the de-doping and doping can be found in Supplementary Note 4. In this regime, there is presumably little

| Table 3 Transport parameter and doping values for different Copper loading | | |
|---|---|---|
| **PEDOT:PSS-Cu-Te** | **$\sigma_{E0}$ (S/cm)** | **η** |
| 0% | 5.47 | 1.62 |
| 10% | 1.78 | 1.28 |
| 50 % | 1.52 | 14.59 |

change in the PEDOT morphology at the organic-inorganic interface, and the increased conductivity and reduced Seebeck are a result of a doping effect associated with additional copper loading. Hence, both 10% and 50% Cu loading samples lie on the $s = 1$ curve with same $\sigma_{E_0}(T)$ values (Fig. 4a). Coupled with other PEDOT-based hybrids (Fig. 4a), we can conclude generally that the energy-dependence of carrier scattering is independent of the type of doping, or indeed the inorganic constituent, contrary to many reports in the literature. This result provides key evidence that carrier transport in hybrid films occurs predominantly through PEDOT itself (to corroborate this, we also performed experiments on Te NWs in an insulating polymer matrix: details can be found in Supplementary Figure 17). Counter-ions and inorganic constituents impact transport mainly via the transport parameter,$\sigma_{E_0}(T)$which can be attributed to morphological/ templating effects in the PEDOT phase. The increase in $\sigma_{E_0}(T)$ in hybrid materials can be qualitatively understood as enhancing the effective mobility of the itinerant carriers within the PEDOT polymer matrix. It is interesting, albeit counter-intuitive, that this can occur as a result of introducing numerous new potential scattering interfaces in the material via addition of inorganic components or secondary phases. However, that introduction of inorganic species can provide templating effects in polymers which lead to structural or behavioural modifications is well-known.

To gain deeper insight into the transport coefficient, $\sigma_{E_0}(T)$, we performed temperature dependent Seebeck and conductivity measurements on these hybrid films. The reduced chemical potential with respect to room temperature value, η/η$_{300K}$, does not change significantly with lowering temperature as shown in Fig. 4b (24%, 35% and 40% for 0, 10 and 50 % Cu loading respectively from room temperature value). The temperature dependence of $\sigma_{E_0}(T)$ can be written as $\sigma_{E0} \propto \exp\left(-\frac{W_\gamma}{k_BT}\right)^\gamma$ where $W_\gamma$ is akin to a hopping energy; while $W_\gamma$ shows a small enhancement with initial Cu loading ($W_\gamma = 0.57$ eV for 0% Cu & 0.69 eV for 10 %), it decreases for 50 % Cu loaded sample (0.28 eV) [Supplementary Table 2] (detailed discussion in Supplementary Note 3, Supplementary Figures 8-9).

In order to test if our hypothesis that charge conduction in hybrid systems occurs mainly through the polymer matrix, we also consider hybrid films constituted from a different (i.e. non-PEDOT) well-established conducting polymer as a matrix, P3HT. We compared, in a similar fashion as above, F$_4$TCNQ vapor-doped P3HT samples[31] (closed circles), Fe((CF$_3$SO$_2$)$_2$N)$_3$ doped P3HT (open squares)[32], highly aligned P3HT with trichlorobenzene (closed squares)[33], to P3HT:Bi$_2$Te$_3$ (closed stars)[34], P3HT:SWCNT (open star) and P3HT:MWCNT (closed triangles) hybrid systems from literature[35–38] and used the Kang-Snyder model (Fig. 5a). These P3HT-based systems lie on the $s = 3$ curve with same $\sigma_{E_0}(T)$ (Table 4). In the case of P3HT:SWCNT (MWCNT) hybrid systems, while the sparse literature data available lies on the $s = 3$ curve with different value of $\sigma_{E_0}$, some do indeed deviate for higher values of conductivity. There is a possibility of strong π-π interactions between CNTs and P3HT, which has been hypothesized to cause deviations from the $s = 3$ curve[39] and is indeed an exciting future avenue for exploration.

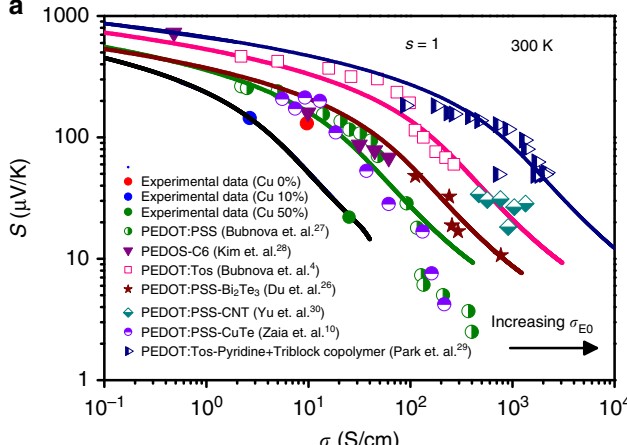

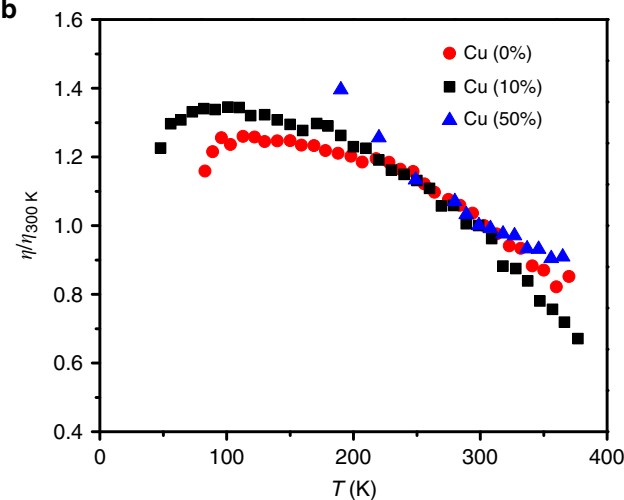

**Fig. 4** Kang-Snyder transport model applied to various PEDOT based composites. **a** Experimental data of Seebeck (S) vs conductiviy (σ) for PEDOT:PSS (half closed green triangles)[27], PEDOS-C6 (closed triangles)[28], PEDOT:Tos (open squares)[4], PEDOT:PSS-Bi$_2$Te$_3$(closed triangles)[26], PEDOT:PSS-CNT[30](half closed spades), PEDOT:PSS-Te(Cu$_x$) NW hybrid system (half closed circles)[10] and PEDOT:Tos-Pyridine (half closed triangles)[29] modeled with s = 1(solid lines). It is seen that, irrespective of the dopant counter-ion used, all hybrid PEDOT-based systems lies on s = 1 curve with different $\sigma_{E_0}$ transport coefficient values, indicating that energy-dependent scattering is not changing in these organic-inorganic hybrid films. **b** Reduced chemical potential, η = (E$_F$-E$_t$/k$_B$T) of the PEDOT:PSS-Te (Cu$_x$) system plotted as a function of temperature for 0% (red closed circle), 10% (black closed squares) and 50 % Cu (closed blue triangles) samples, respectively. In line with expectations from the Kang-Snyder model, the reduced chemical potential only changes < 30% over a large temperature range in the samples and the change does not depend upon the Cu loading. The data is normalized with respect to the doping level at 300 K (η$_{300K}$)

The conclusion, in the case of P3HT:Bi$_2$Te$_3$, is identical to that for the PEDOT based films wherein the energy dependence of the scattering in the hybrid films is similar to that of pristine polymeric films. Figure 5b shows power factor (S$^2$σ) for the PEDOT and P3HT based systems. While for PEDOT based hybrids, the power factor shows an optimum for different $\sigma_{E_0}(T)$ values, it increases continuously for P3HT based hybrids. The value of power factor is hypothesized to be lower in P3HT based systems because of a low value of $\sigma_{E_0}$ due to side alkyl side-chains; although alkyl chains help to make these solution processable, it

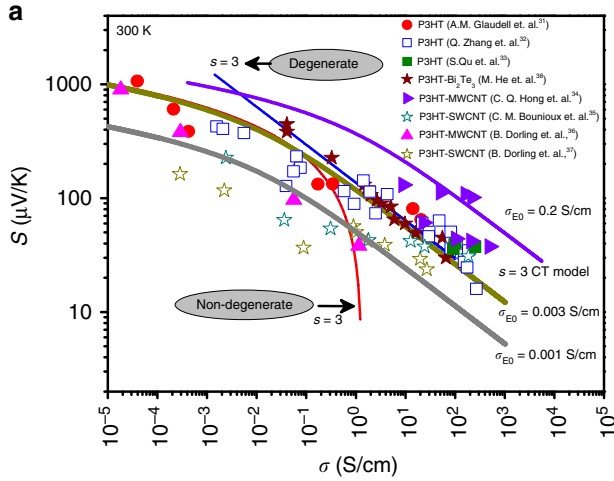

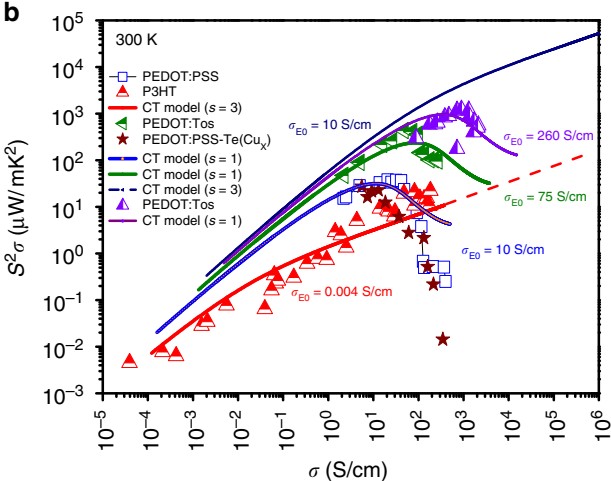

**Table 4 Transport parameter for PEDOT and P3HT based pure polymer and hybrid films**

| Material | $\sigma_{E0}$ (S/cm) |
|---|---|
| PEDOT:PSS-C6 | 8.5 |
| PEDOT:PSS (electrochemical) | 8.5 |
| PEDOT:Tos | 61 |
| PEDOT:Tos + Pyridine | 260 |
| PEDOT:PSS-Bi$_2$Te$_3$ | 14 |
| P3HT:Bi$_2$Te$_3$ | 0.003 |
| P3HT:MCNT | 0.2, 0.001 |
| P3HT:SWCNT | 0.001 |

**Fig. 5** Kang-Snyder model applied to various P3HT-inorganic composites. **a** The electrical conductivity vs Seebeck coefficient data of F$_4$TCNQ doped P3HT (closed circles)[31], Fe((CF$_3$SO$_2$)$_2$N)$_3$ doped P3HT (open squares)[32], highly aligned P3HT with trichlorobenzene (closed squares)[33], P3HT: MWCNT (closed triangles)[35,36], P3HT:SWCNT (open star)[37,38] and P3HT: Bi$_2$Te$_3$ (closed stars)[34] hybrid systems. It is seen that experimental data lies on s = 3 curve, again consistently identical for the hybrid and the pure polymeric systems. **b** Comparison of power factor of PEDOT (s = 1) and P3HT (s = 3) based hybrids. Targeting higher $\sigma_{E_0} \sim 10$ S/cm in a s = 3 polymer can push power factors of hybrid materials towards values comparable to inorganic thermoelectric materials

degrades the alignment and orientation of the P3HT polymer chains[40,41]. Hypothetically, if a P3HT-based hybrid film with $\sigma_{E_0}$ approaching 10 S/cm is manufactured, the power factor would be as high as 10 mW/mK$^2$. Therefore, it is clear that, so far, the key to high thermoelectric performance in these complex hybrid systems has been the advantage gained by physical interfacial interactions and exploiting polymeric templating effects capable of enhancing carrier transport in the organic phase, rather than modifying the energy dependence of scattering.

## Discussion

In conclusion, by combining experiment, first principles calculations, and molecular dynamics, we show that the high thermoelectric performance achieved in PEDOT:PSS-CuTe nanowires is driven primarily by thermoelectric transport in the organic phase. Contrary to previous understanding, this transport is enhanced due to a physical templating effect at the organic-inorganic interface accompanied by charge transfer induced de-doping at the interface, rather than cooperative interfacial transport or modification of the energy dependence of scattering. Significantly, we apply the Kang-Snyder charge transfer model to a wide variety of organic-inorganic hybrids and demonstrate that it provides an effective framework to describe composite materials. Pairing our experimental data with results from literature, we demonstrate that the key to high performing hybrid materials lies in the energy-independent transport coefficient, $\sigma_{E_0}(T)$. This provides a general result suggesting that the role of energy dependent scattering in hybrid materials has been systematically overestimated. Instead, transport in most hybrid systems can be understood within the context of the individual components, with enhancements arising from physical interactions such as templating. In summary, this work lays out a clear framework for development of next generation soft thermoelectrics: leverage upon stronger energy dependent scattering (s = 3) polymers, enhance chemical interactions between inorganic and organic constituents, and create architectures and templates emphasizing interfacial design capable of high-conductivity domains in the organic phase that enhance mobility of charge carries. These are three well-defined routes that can be impactful in the near future.

## Methods

**Synthesis and Characterization.** Synthesis of PEDOT:PSS-Te NWs and PEDOT: PSS-Te(Cu$_x$) NWs closely followed previous methods[10,11]. All steps in the procedure are carried out in aqueous solution in the presence of air, with high reproducibility over many separate experiments. As shown in our previous report, during conversion to PEDOT:PSS-Te(Cu$_x$) NWs, mobile copper ions penetrate the PEDOT:PSS surface layer to react with the Te core and form isolated alloy domains of Cu$_{1.75}$Te[10]. During this process, the nanowires undergo a transition from rigid rods (Fig. 1a) to curved wires, with alloy domains appearing at 'kinked' portions of the wires (Fig. 1b). The resulting nanostructures are Te-Cu$_{1.75}$Te heterowires. The extent of copper loading in each sample was directly measured using inductively coupled plasma optical emission spectroscopy (ICP-OES). After synthesis, x-ray diffraction (XRD), scanning electron microscopy (SEM), and x-ray photoelectron spectroscopy (XPS) were used to confirm the material structure and properties. Representative data can be found in the Supplementary Information (Supplementary Figures 12-13) and are consistent with our previous report on these materials[10,11].

**MD Simulations.** In order to investigate the morphology and configuration of PEDOT:PSS on Te and Cu$_{1.75}$Te NW surfaces, a 27 × 20 supercell (12.0 × 11.8 nm$^2$) of Te [100] surface with 4947 Te atoms and a 6 × 28 supercell (10.8 × 11.1 nm$^2$) of Cu$_{1.75}$Te [010] surface with 9073 atoms (Cu$_{5787}$Te$_{3286}$) are constructed, respectively. Ten and eight layers of atoms were used for the surface thickness which corresponds to 13.36 Å and 12.07 Å for Te and Cu$_{1.75}$Te, respectively. Molar ratio of polymers were determined as 1:2 for PEDOT:PSS in simulations according to experimental results[42].

Canonical ensemble-molecular dynamics (NVT-MD) are used within the framework of Forcite Plus package of Materials Studio[43]. Ten annealing cycles between 300 K and 1300 K were carried out for equilibrium followed by 5000 steps smart minimization. The total annealing time is at least 5 ns with 0.5 fs time step and Nosé-Hoover Thermostat method is adopted for temperature control. Condensed-phase Optimized Molecular Potentials for Atomistic Simulation

Studies (COMPASS)[44] is used to evaluate the atomic forces[45,46]. A summation method of non-bonded electrostatic forces controlled by Ewald[47] and van der Waals forces by "atom-based" is employed in periodic cells. 11 Å atomic cut-off distance was used for vdW interactions. Three different cubic amorphous cells were prepared including five chains of $EDOT_{18}$ and $SS_{36}$ mixture for PEDOT:PSS simulations (Fig. 1e, f, Supplementary Figure 1), 20 chains of $EDOT_{18}$ for pristine PEDOT simulations and 10 chains of $SS_{36}$ for pristine PSS simulations on Te and $Cu_{1.75}Te$ surfaces respectively (Supplementary Figures 2-3). Each simulation was repeated at least three times with simulated annealing protocol for different amorphous polymers to investigate structural changes.

Six $EDOT_{12}$ and three $SS_{24}$ oligomers were used to calculate interaction energies with the Te and $Cu_{1.75}Te$ NW surfaces (Supplementary Figure 4). Similarly, six $EDOT_{12}$ oligomers distributed randomly onto the NW surface was used to study directional preference and self-assembly of chains on surfaces (Supplementary Figure 5).

**DFT Calculations**. Based on the experimental results, a Te [100] surface[48] is cut with six subtract layers under it and a 20 Å vacuum slab above it to ensure no interlayer interaction (Supplementary Figure 14). The simulated XRD diffraction pattern of this model agrees exactly with the experimental TEM and SEM images (Supplementary Figure 15)[25,49–51]. A $3 \times 6$ ($13.2 \times 35.5$ Å$^2$) supercell with a PEDOT hexamer aligned along the Te NW growth direction and a $6 \times 3$ ($26.4 \times 17.7$ Å$^2$) supercell with a PEDOT hexamer a perpendicular to the Te NW growth direction are prepared, respectively. Similarly, a $Cu_{1.75}Te$[52] [010] surface cut and a $2 \times 3$ ($43.4 \times 12.5$ Å$^2$) supercell with 20 Å vacuum above are prepared respectively followed the same method (Supplementary Figure 16). Adsorption energy, density of states (DOS), electrostatic potential surface and electron density differences between surface and PEDOT are calculated based on Density Functional Theory[53] within the framework of plane-wave implemented in the Cambridge Serial Total Package (CASTEP)[54]. Tkatchenko-Scheffler (TS)[55] scheme dispersion corrected Perdew–Burke–Ernzerhof functional[56] with van der Waals consideration (PBE-D) are adopted as exchange-correlations, and generated on the fly (OTFG) ultrasoft potentials are used to describe interactions with a cutoff energy, 490 eV for Te for 571 eV for $Cu_{1.75}Te$. Total energy convergence is $1 \times 10^{-6}$ eV/atom and the force on the atom is 0.03 eV. The maximum stress is 0.05 Gpa and the maximal displacement is 0.01 angstrom. BFGS algorithm is used for geometry optimization and surface relaxation. The adsorption energy was calculated according to the equation: $E_{[ads]} = E_{[total]} - (E_{[monomer]} + E_{[NW]})$; here, $E_{[ads]}$, $E_{[total]}$, $E_{[monomer]}$ and $E_{[NW]}$ represent the adsorption energy, the total energy of a single monomer/oligomer adsorbed on the Te surface, the energy of isolated monomer/oligomer and the energy of Te-based nanowires respectively. Molecular properties such as electrostatic potential surfaces and highest occupied molecular orbitals are calculated by using DMOL3 software at the same level of DFT functional[57].

## Data availability
The datasets generated during and/or analysed during the current study are available from the corresponding author on reasonable request.

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

## Acknowledgements

This work was partially performed at the Molecular Foundry, Lawrence Berkeley National Laboratory, and was supported by the Department of Energy, Office of Science, Office of Basic Energy Sciences, Scientific User Facilities Division of the U.S. Department of Energy under Contract No. DE-AC02-05CH11231. This work was also partially performed at the Agency for Science, Technology, and Research, supported by the Science and Energy Research Council under the Pharos grants 1527200018 and 1527200024. EWZ gratefully acknowledges the National Science Foundation for fellowship support under the National Science Foundation Graduate Research Fellowship Program.

## Author contributions

PK and DMR performed thermoelectric modeling using Kang Snyder model and temperature dependent electrical and thermoelectric measurements. EWZ synthesized all PEDOT-Te(Cu$_x$) materials and performed physical characterization (TEM, XPS, XRD). EY and SWY performed MD simulations and DFT calculations. JJU and KH conceived and directed the project. PK, EWZ, and KH wrote the paper with input from all authors. All authors reviewed the final paper. EY, SWY, DMR, JJU, KH.

## Additional information

**Competing interests:** The authors declare no competing interests.

