## [Peer Review File · Nature Communications]

Reviewers' comments:

Reviewer #1 (Remarks to the Author):

The manuscript by Kumar, et al. presents data argues that the observed increase in the TE performance of PEDOT:PSS-(Cu)Te NW hybrids is due primarily to the alignment of PEDOT segments brought about by a templating effect. This conclusion is in contrast to previous reports where energy filtering was suspected as the origin of the enhancement. The hypothesis of the NW templating effect is supported by molecular dynamics and DFT simulations from a theoretical standpoint, and through fitting data with the generalized Kang-Snyder transport model. The Kang-Snyder transport model is also applied to a series of other polymers and polymer:NW hybrids and shown to yield reasonable fits. The molecular dynamics and DFT calculations are indeed interesting and important for understanding transport in these systems. However, the large deviations (up to an order of magnitude) between experimental data and the Kang-Snyder transport model fits makes it hard to imagine that this model could show whether or not small changes in the Seebeck coefficient and electrical conductivity are due to energy filtering. Overall, the paper is well written and may make a significant contribution of broad interest to the organic and hybrid thermoelectric community if the following points can be adequately addressed. The paper should be reconsidered for publication in Nature Communications following revisions.

1) My main concern with the manuscript is in the relatively loose fits afforded by the Kang-Snyder transport model and the ability to definitively assign subtle deviations in transport behavior based on this model. For example, in this manuscript, as well as in the Kang, Snyder Nature Materials 2017 paper, the fits are shown on a logarithmic scale and the data points are up to an order of magnitude off the model fit line. The fact that this model fits so many polymers to within an order or magnitude or so indeed lends strong support for its generalizability, but it is difficult to identify subtle details of charge transport. For example, the original claim of energy filtering in these PEDOT:PSS-Te(Cu) hybrids is based primarily on a $\leq 10\%$ increase in the Seebeck coefficient. This 10% increase in the Seebeck coefficient will be completely buried in the fits afforded by the Kang-Snyder transport model. The authors need to find some way (experimental and/or theoretical) to show that the amount of energy filtering required for the 10% increase in the Seebeck coefficient would indeed lead to deviation from the $s=1$ fit, given the large fit tolerance.

2) Why does $s=1$ for the hybrid necessarily imply that transport occurs predominantly through the PEDOT? Could not $s=1$ for the pure nanowires as well, in which case s may not change regardless of whether transport is through the nanowires or PEDOT? The authors need to provide additional arguments and/or experimental data to support that $s=1$ implies charge transport is dominated by the polymer.

3) The authors state on lines 211-212 that the thermoelectric behavior in p-type PEDOT-Te hybrids is dominated by transport through the organic PEDOT matrix. This conclusion is reached based on negligible ground state charge transfer between the PEDOT and Te, as also discussed on lines 172-179. However, high interfacial charge transfer rates may still be obtained in the absence of ground state charge transfer. The authors' statements need to be further explained and supported.

4) The authors should include the equation for η in the text and should state that it is actually the reduced chemical potential. The reduced chemical potential will largely depend on doping, but it is not solely determined by the number of charge-carriers as the authors current description implies. The authors need to also state how this reduced chemical potential was determined. Also, it would be helpful if the authors suggest how a small amount of Cu may decrease η .

5) It is a big stretch to say that the P3HT-CNT data lies on the purple curve shown in Figure 5a. I recommend removing this data series and seeing if a similar set of data in the literature may provide data over a larger S and conductivity range that can be used instead.

6) More details on the measurement geometry for the Seebeck coefficient measurements need to be given. Improper measurement geometries can lead to large errors in the Seebeck coefficient, see van Reenen and Kemerink, *Organic Electronics*, 2014, 15, 2250. More details on the temp dependent electrical conductivity and Seebeck measurements should also be provided – specifically electrode geometries and distances.

Additional points:

In the caption of Figure 4, the authors state that the doping level (again, doping level should not be used) only changes <30%. This statement is misleading since the value appears to reach a maximum of 1.4 and a minimum of 0.7. Statement needs to be reworded and/or the value changed.

It seems that two different PEDOT-Te interaction energies are reported, 160 kcal/mol on line 148 and 88 kcal/mol on line 159-160. These should either be corrected or the reasons for these differences clarified.

At one point the authors state that they are below the percolation threshold in the PEDOT:PSS-Te(Cu) hybrids, but from the SEM images this does not appear to be the case. It would be helpful if the authors could state both the volume and weight % of the NWs in the films, as well as the percolation threshold.

The authors should further explain where the various lines presented in Figure 3b originate from (degenerate, non-degenerate, and CT). Presumably CT stands for charge transport model and the authors are using this to denote that this is the complete model, but this should be listed.

The sentence “organic semiconductors are successfully modeled as semiconductors” is odd – they are semiconductors and of course can be modelled as such. Is this meaning to say that they are modeled using theories originally developed for inorganic semiconductors?

Reviewer #2 (Remarks to the Author):

This paper presents detailed theoretical and experimental results on thermoelectric properties of PEDOT:PSS-Te/CuTe nanowire. It concludes that there is no modification of the energy dependence of scattering or charge transfer between polymer and the inorganic nanowire. The Kang-Snyder equation was used to explain the transport properties of the hybrid material, based on the assumption that the transport properties are dominated by the polymers. The work is carefully done and the conclusions would be informative for the researchers in the field of hybrid organic-inorganic thermoelectric materials.

It is already known that Tellurium has very high thermoelectric performance, [Nat. Commun. 7, 10287 (2016)], so does it contribute to the thermoelectric performance of the hybrid material? Cu_xTe also has very high power factor (reference [25]). How to exclude its effect from the total thermoelectric performance of the hybrid material. If there is no charge transfer between the polymer and the Te/Cu_{1.75}Te nanowire, why the Seebeck coefficients of PEDOT:PSS-TE and PEDOT:PSS-Cu_{1.75}Te show a big difference (line 238). It is not clear how the loading of copper increases the carrier concentration of the polymer.

Pure organic PEDOT:PSS has already achieved a very high ZT value of 0.25-0.42. [Nat. Mater. 10, pages 429–433 (2011) & Nat. Mater. 12, pages 719–723 (2013)] However, the current work is mainly based on a hybrid PEDOT:PSS-Te composite with ZT=0.1. The multiphase approach does not show advantage over the pure organic materials on the thermoelectric performance.

The paper provides a deep understanding on the transport properties of PSS-PEDOT/Te nanowire that has been previously reported. It is quite strange that Figure 3a in this paper is very close to the Figure 3a in the previous work of the authors (Reference [10]). Future directions for hybrid organic-inorganic thermoelectric materials have been pointed out, but it seems sufficiently striking insights are lacking.

The manuscript by Kumar, et al. presents data argues that the observed increase in the TE performance of PEDOT:PSS-(Cu)Te NW hybrids is due primarily to the alignment of PEDOT segments brought about by a templating effect. This conclusion is in contrast to previous reports where energy filtering was suspected as the origin of the enhancement. The hypothesis of the NW templating effect is supported by molecular dynamics and DFT simulations from a theoretical standpoint, and through fitting data with the generalized Kang-Snyder transport model. The Kang-Snyder transport model is also applied to a series of other polymers and polymer:NW hybrids and shown to yield reasonable fits. The molecular dynamics and DFT calculations are indeed interesting and important for understanding transport in these systems. However, the large deviations (up to an order of magnitude) between experimental data and the Kang-Snyder transport model fits makes it hard to imagine that this model could show whether or not small changes in the Seebeck coefficient and electrical conductivity are due to energy filtering. Overall, the paper is well written and may make a significant contribution of broad interest to the organic and hybrid thermoelectric community if the following points can be adequately addressed. The paper should be reconsidered for publication in Nature Communications following revisions.

1) My main concern with the manuscript is in the relatively loose fits afforded by the Kang-Snyder transport model and the ability to definitively assign subtle deviations in transport behavior based on this model. For example, in this manuscript, as well as in the Kang, Snyder Nature Materials 2017 paper, the fits are shown on a logarithmic scale and the data points are up to an order of magnitude off the model fit line. The fact that this model fits so many polymers to within an order or magnitude or so indeed lends strong support for its generalizability, but it is difficult to identify subtle details of charge transport. For example, the original claim of energy filtering in these PEDOT:PSS-Te(Cu) hybrids is based primarily on a $\leq 10\%$ increase in the Seebeck coefficient. This 10% increase in the Seebeck coefficient will be completely buried in the fits afforded by the Kang-Snyder transport model. The authors need to find some way (experimental and/or theoretical) to show that the amount of energy filtering required for the 10% increase in the Seebeck coefficient would indeed lead to deviation from the $s=1$ fit, given the large fit tolerance.

Response- We thank the Reviewer for their insightful comment. We agree with the Reviewer that the Kang-Snyder model was developed to explain the relationship between conductivity and

Seebeck for different polymers over a large range of conductivities, and therefore sacrifices some fine-grained predictive power. We now mention this explicitly in the revised manuscript on line 20, page 14 as ‘while the Kang-Snyder model is a powerful tool to understand overall trends over a large range of conductivity and Seebeck, there is not enough sensitivity to the fitting parameters, prohibiting its usage to discriminate effects such as electron-filtering’.

Hence, in order to strengthen our argument that there is no *additional* energy-dependent filtering, we provide extensive theoretical evidence to support our understanding of the mechanisms behind the non-monotonic trends observed in the thermoelectric properties of the PEDOT:PSS-Te/CuTe system. We have analyzed the effect of Copper loading and interfacial charge states even more carefully for the revised manuscript. This also follows from the 3rd question of the Reviewer, pertaining to interfacial charge transfer at the inorganic-organic interface. We specifically looked for charge transfer in new DFT simulations and studied their effects on doping/de-doping of the hybrid material. There are three effects that control thermoelectric transport in this hybrid system:

1. Templating of the conducting PEDOT due to the inorganic (Te/CuTe) nanowire surface, (*column 1 in the table below*) is defined as self-alignment of PEDOT chains on Te and CuTe atomic surfaces as shown in Fig. S1. Here, the carrier mobility is enhanced and therefore the conductivity increases. There is no change in carrier concentration.
2. De-doping due to charge transfer at this organic- inorganic interface. We have performed extensive DFT simulations of the inorganic-organic interface to demonstrate that there is indeed charge transfer from the inorganic Te/CuTe to the PEDOT layer as shown by the blue color in Fig. 2 in the revised manuscript. The electron transfer from Te/CuTe to PEDOT can be understood as de-doping of PEDOT (less holes within PEDOT chains, *column 2 in the table below*). Please see the response to the Reviewer’s third question for in-depth discussion of this effect.
3. As described in the original manuscript, Cu⁺ ion loading results in additional charge carriers being introduced into the conducting PEDOT chains, accompanied with a reduction in the templating effect (*column 3 in the table below*).

Table R1. Summary of sequential addition of Te and then Cu_{1.75}Te inorganic phases to PEDOT:PSS and their role on thermoelectric transport

	Templating effect	De-doping due to charge transfer at organic-inorganic interface (CT)	Cu loading	Total Conductivity and Seebeck coefficient	Discussion
PEDOT:PSS	X	X	X	X	X
PEDOT:PSS-Te	$\mu(\uparrow)$ (strong)	$n_{de-doping}(\downarrow)$ $S_{de-doping}(\uparrow)$ (strong)	X	$\sigma(\uparrow) = n_{de-doping}(\downarrow) \times \mu(\uparrow)$ $S(\uparrow) \propto \frac{1}{n_{de-doping}(\downarrow)}$	Conductivity increases due to templating; Seebeck increases due to charge transfer induced de-doping – higher than PEDOT:PSS
PEDOT:PSS-Cu _{1.75} Te (low Cu loading)	$\mu(\downarrow)$ (weak)	$n_{de-doping}(\downarrow)$ $S_{de-doping}(\uparrow)$ (strong)	$n_{doping}(\uparrow)$ $S_{doping}(\downarrow)$ (weak)	$\sigma(\downarrow) = \mu(\downarrow) \times \{n_{de-doping}(\downarrow) + n_{doping}(\uparrow)\}$ $S(\uparrow) \propto \frac{1}{\{n_{de-doping}(\downarrow) + n_{doping}(\uparrow)\}}$	Conductivity decreases due to further drop in templating; Seebeck increases due to stronger de-doping
PEDOT:PSS-Cu _{1.75} Te (High Cu loading)	$\mu(\downarrow)$ (strong)	$n_{de-doping}(\downarrow)$ $S_{de-doping}(\uparrow)$ (weak)	$n_{doping}(\uparrow)$ $S_{doping}(\downarrow)$ (strong)	$\sigma(\uparrow) = \mu(\downarrow) \times \{n_{de-doping}(\downarrow) + n_{doping}(\uparrow)\}$ $S(\downarrow) \propto \frac{1}{\{n_{de-doping}(\downarrow) + n_{doping}(\uparrow)\}}$	Conductivity increases due to strong Cu ⁺ induced doping; Seebeck decreases due to stronger Cu ⁺ induced doping

In the revised manuscript on Page 7, line 13, we describe in detail that Te/Cu_{1.75}Te nanowires are coated with ~2nm thin PEDOT layer. MD simulations show that the PEDOT chains align nicely on the Te surface and this alignment fades as the curved Cu_{1.75}Te phase grows within the Te nanowires. Detailed discussion of how these three effects interact to result in the observed thermoelectric trends for the hybrid system follows in the next two paragraphs.

The non-monotonic thermoelectric trends are explained by the interaction of multiple effects. Upon addition of Cu, the curved Cu_{1.75}Te phase within the Te nanowire expands and weakens the templating and de-doping effects. This would cause a decrease in conductivity and a small increase in Seebeck coefficient compare to pure PEDOT/Te, the effect of which is strongest in the low Cu loading regime. At the same time, as mentioned in the revised manuscript on page 15, line 20, not all Cu ions interact with the Te nanowire; instead, some of them dope the PEDOT chains. This doping effect is understood to cause an increase in the conductivity and decrease in the Seebeck

coefficient with increasing Cu content, especially apparent in the high Cu loading regime. These conclusions are supported theoretically via MD simulations and DFT calculations.

To understand the thermoelectric properties of PEDOT:PSS-Te, it is important to consider the first two effects listed above. MD simulations show that the PEDOT aligns on the Te surface (i.e. a strong templating is observed), which enhances the mobility of the carriers in the PEDOT chains. On the other hand, due to interfacial charge transfer, de-doping occurs giving rise to a large Seebeck coefficient (note that in the Kang-Snyder model, the Seebeck does not depend upon the σ_{E_0} value and instead only on the carrier concentration via the reduced chemical potential, η). The enhancement in mobility compensates for the reduction in carrier concentration that occurs in PEDOT chains relative to the pristine polymer, and therefore both the Seebeck coefficient and electrical conductivity are enhanced. In order to understand this, consider that in order for the Seebeck to be enhanced from ~ 10 $\mu\text{V}/\text{K}$ (pristine polymer) to ~ 200 $\mu\text{V}/\text{K}$ (PEDOT:PSS-Te composite), a carrier concentration decrease of one order of magnitude is required – on the other hand the mobility is expected to increase by two orders of magnitude.

To elucidate the mechanism behind the non-monotonic trend in thermoelectric properties in the full PEDOT:PSS-Te/CuTe system, a combination of all three effects described above is required. First, we have performed new DFT simulations that show that the charge transfer between PEDOT and $\text{Cu}_{1.75}\text{Te}$ is stronger than that for PEDOT and Te, resulting in a stronger de-doping effect (reduction in carrier concentration) upon addition of Cu to the system, especially in the low Cu loading regime. However, due to the curved nature of the $\text{Cu}_{1.75}\text{Te}$ nanowires, the de-doping effect on PEDOT/ $\text{Cu}_{1.75}\text{Te}$ surface is not as effective as PEDOT/Te, and therefore the Seebeck coefficient shows only a small enhancement in this regime (200 to 213 $\mu\text{V}/\text{K}$). Second, as discussed above, the growth of $\text{Cu}_{1.75}\text{Te}$ surface within the Te nanowire weakens the templating, thus decreasing the carrier mobility in the organic phase. Third, Cu loading also dopes the PEDOT chains via ions present in the polymer phase. In the low Cu loading regime, Cu doping (effect 3) is inefficient compared to the charge transfer between PEDOT-Te/ $\text{Cu}_{1.75}\text{Te}$ (effect 2) and therefore, the Seebeck is enhanced. In the case of conductivity, both de-doping (carrier reduction) and weak templating (mobility reduction) will dominate, which is why decreased conductivity is observed in this range. As the Cu loading is increased, the $\text{Cu}_{1.75}\text{Te}$ surface with Te nanowire

expands, burying the effects of templating and charge transfer induced de-doping, and instead, the doping of PEDOT with Cu (effect 3) will dominate as is observed experimentally. Combined with the results of fitting using the Kang-Snyder model, our extensive theoretical analysis strongly suggests that this intricate interplay of templating, Cu doping and charge transfer at the interfacial polymer state is responsible for the observed thermoelectric properties, and not a change in the energy-dependence of scattering as was originally proposed.

The paragraph is added in revised manuscript on page 14, line 22:

“Note, however, that while the Kang-Snyder model captures large trends in the S vs σ curve, small changes such as electron filtering cannot be isolated. Hence, in order to understand the non-monotonic trend in the Seebeck and conductivity, we study in detail the effect of (de-)doping and templating on the hybrid system (Table I). Combining our experimental and theoretical results, we conclude that the complex thermoelectric trends of these hybrid films are dictated by the interaction of several effects. First, as suggested by extensive MD simulations, upon the formation of PEDOT:PSS-Te NWs, there is a templating effect for PEDOT moieties on the inorganic surface. This phenomenon results in an increase of hole mobility in the interfacial polymer, increasing the electrical conductivity of the PEDOT:PSS-Te composite relative to the pristine polymer. This templating effect is weakened by the addition of Cu, which disrupts the inorganic surfaces and produces “kinked” inorganic morphologies. Secondly, detailed DFT calculations are indicative of charge transfer between the organic and inorganic phases, resulting in a de-doping effect of the p-type PEDOT chains (Table S.1). In the low Cu loading regime, this de-doping effect is relatively strong, and contributes directly to the increased Seebeck coefficient and moderately decreased electronic conductivity observed here. This is contrary to previous hypotheses that a change in the energy dependence of carrier scattering is solely responsible for the non-monotonic thermoelectric trends observed in this range.

Upon further addition of Cu, a third effect emerges; Cu loading above 10% is associated with an increase in η , with only a nominal change in the σ_{E_0} value (Table II). This trend indicates that the addition of Cu introduces carriers into the film and modifies the transport through a doping channel. Previous reports on this material system have suggested that positively charged Cu ions, in addition to reacting with the inorganic phase, also remain in the PEDOT phase as

ionic species. These remaining Cu ions likely interact with the PEDOT chains to increase the carrier concentration in the organic phase. This effect dominates at high Cu loading, which is associated with a strong increase in the reduced chemical potential. Note that $s=2$ and $s=3$ do not fit the experimental data for any value of the transport coefficient, $\sigma_{E_0}(T)$ (Figure 3(b) is plotted on log-log scale). While, for a fixed σ_{E_0} , η is modulated by charge redistribution between the organic and inorganic phases and doping from Cu ions, only a change in morphology (templating, or kinked surfaces) can change σ_{E_0} . ”

The Table 2 is added in revised manuscript on page 21 (same as Table R1 above).

2) Why does $s=1$ for the hybrid necessarily imply that transport occurs predominantly through the PEDOT? Could not $s=1$ for the pure nanowires as well, in which case s may not change regardless of whether transport is through the nanowires or PEDOT? The authors need to provide additional arguments and/or experimental data to support that $s=1$ implies charge transport is dominated by the polymer.

We thank the Reviewer for their insightful question. The $s=1$ fit is a classic indicator for transport in PEDOT relative to other semiconducting polymers, and for consistency with the organic thermoelectrics literature, this point is stressed. However, this is not sufficient to prove that transport occurs predominantly through the PEDOT phase in organic-inorganic composite films. Thus, we have performed additional experimentation, which, when combined with the data presented in our paper, provides additional support for this claim. We further elaborate upon this conclusion in response to question number 3 below, where we focus on charge transfer between the inorganic and organic phases.

In addition to the PEDOT:PSS-Te/CuTe NWs, we have synthesized Te/CuTe NWs in an insulating polymer matrix (polyvinylpyrrolidone (PVP), 55 kDa), keeping the polymer-to-inorganic ratio in each film roughly the same by mass as for the PEDOT composites. We performed electrical and thermoelectric transport measurements, and modeling according to the Kang-Snyder model (added as Fig S17 in the Supplementary Information; appended below). The results of this experiment shows that the transport properties in these composite films cannot be explained as the result of

transport in a percolated network of inorganic nanostructures, instead suggesting a key role for the organic component.

Fig. S17. Experimental data of Seebeck (S) vs conductivity (σ) for Te/CuTe nanowire embedded in insulating polymer matrix (closed square), PEDOT:PSS-Te(Cu_x) NW hybrid system (closed circles) and bulk Te (closed star) modelled with $s=1$ (solid lines). It is seen Seebeck vs conductivity data of PEDOT-based hybrid system and bulk Te lies on $s=1$ curve with different σ_{E_0} transport coefficient values. In case of Te/CuTe nanowire embedded in insulating polymer matrix, the Seebeck vs conductivity data lies on all possible s values ($s=1, 2,$ and 3) and it is difficult to distinguish that which ‘ s ’ is true for the system.

If conductivity in these hybrid systems were described by transport through a percolated inorganic network, then one would expect the measured conductivity to be independent of the organic matrix. However, for the PVP-Te NW system, the electrical conductivity is drastically reduced relative to the PEDOT-Te NW system or even to bulk Te; the conductivity of PVP-Te NWs is 0.015 S/cm, which is approximately 1000 times lower than that of bulk Te. Even at maximum Cu loading, the maximum conductivity measured for PVP-Te/CuTe NWs is 0.7 S/cm, which is also 1000 times lower than bulk $\text{Cu}_{1.75}\text{Te}$ (B. A. Mansour et. al., *Thin Solid Films*, 247, 112 (1994)) and $\text{Cu}_{1.75}\text{Te}$ nanowires (C. Zhou et. al., *ACS Appl. Mater. Interfaces*, 2015, 7, 21015 (2015)). This low conductivity in the PVP-Te/CuTe supports our conclusion that the organic component dominates transport in our PEDOT-based hybrid films. To investigate this phenomenon further,

we have plotted the transport properties for PVP-Te/CuTe NW, PEDOT-Te/CuTe NW, and bulk Te in Fig S17 below, and fit these data using the Kang-Snyder model. Upon applying the model, we find that, while bulk Te shows $s=1$ dependence with large values of σ_{E_0} , it is impossible to distinguish which s value is true for our Te/CuTe NW in insulating polymer matrix, unlike the clear demarcation in the Te/CuTe-PEDOT system. It is well known that scattering of charges is dominated by acoustic phonon scattering at room temperature for inorganic semiconductors which corresponds to $s=1$ ($r=-0.5$).¹ Therefore, $s=1$ dependence is well-understood in bulk Te (green stars). Hence, if the charge transport was dominated through a percolation pathway through the semiconducting nanostructures, we should expect $s=1$ for all inorganic-organic hybrid systems. However, as described in the original manuscript, this is not the case for the majority of non-PEDOT based hybrid systems. For example, in P3HT based hybrid systems (Fig. 5(a) in the revised manuscript), $s=3$ dependence is conclusively observed, confirming that transport in the polymer matrix is indeed the key factor in the overall film thermoelectric properties.

3) The authors state on lines 211-212 that the thermoelectric behavior in p-type PEDOT-Te hybrids is dominated by transport through the organic PEDOT matrix. This conclusion is reached based on negligible ground state charge transfer between the PEDOT and Te, as also discussed on lines 172-179. However, high interfacial charge transfer rates may still be obtained in the absence of ground state charge transfer. The authors' statements need to be further explained and supported.

Response: We thank the Reviewer for the helpful comments. We have performed additional calculations to extensively consider charge redistribution at the inorganic-organic interface, including interfacial state transfer. The results provide further insight into the physics of carrier transport in the system, and corroborate our conclusions that the thermoelectric behavior in these hybrid systems is dominated by charge transport in the organic phase. We also supplement the evidence provided for the physical nature of the interaction between the PEDOT chains and the inorganic surface.

Using a Mulliken charge analysis, charge transfer rates were calculated for PEDOT hexamers on both Te and Cu_{1.75}Te surfaces. Moreover, pristine state (neutral PEDOT) and bipolaronic/charged

¹ D. A. Neamen, "Semiconductor Physics And Devices" 3rd edition, Macgraw Hill (2003).

PEDOT conditions were considered as the surface PEDOT structures. For the bipolaron case, a +1 charge is added for every three monomers, as predicted experimentally (G. Zotti, S. Zecchin, and G. Schiavon, F. Louwet and, L. Groenendaal, X. Crispin, W. Osikowicz, W. Salaneck and M. Fahlman, *Macromolecules*, 2003, 36, 3337–3344.). Charge transfer rates are then calculated per monomer (see Table 1 and Table S1). Note that a negative quantity here represents electron transfer from the inorganic surface to the organic PEDOT chains (i.e. hole transfer from the organic PEDOT chain to the inorganic surface). As a result of these calculations, it is evident that charge transfer does indeed take place between the organic and inorganic phases. In every case, this charge transfer provides a strong de-doping effect of holes in the p-type PEDOT chains. This de-doping plays a key role in understanding the thermoelectric trends in these hybrid systems, as is discussed in depth above in response to the Reviewer’s first question.

A new paragraph is added to the revised manuscript on page 8, line 20:

“The maximum charge transfer rates from the inorganic surface to the first layer of (neutral) PEDOT chains on the surface were determined to be -0.078 and -0.144 for Te and Cu_{1.75}Te, respectively. Charge transfer rates are higher for the charged (EDOT₆)⁺² bipolaron, calculated to be -0.186 and -0.239 for Te and Cu_{1.75}Te, respectively (Table S1). Note that a negative quantity here represents electron transfer from the inorganic surface to the organic PEDOT chains (i.e. hole transfer from the organic PEDOT chain to the inorganic surface). In every case, this charge transfer provides a de-doping effect of holes in the p-type PEDOT chains, which plays a key role in understanding the thermoelectric trends in these hybrid systems (Table 1). This de-doping effect is stronger for the doped PEDOT bipolaron compared to pristine PEDOT chains and also stronger for PEDOT on the Cu_{1.75}Te surface compared to PEDOT on the Te surface. The charge transfer and de-doping effect is only observed for the first two layer of PEDOT chains and vanishes for higher distances.”

De-doping due to interfacial charge transfer:

A new table has been added to the SI:

Table S1. Charge transfer and de-doping effect for the pristine and doped PEDOT chains on Te and Cu_{1.75}Te surfaces for the geometry optimized structures.

	electron/monomer (Mulliken)	De-doping effect* (electron/cm ³)
Neutral PEDOT ₆ on Te	-0.078 for 3.6 Å (-0.016 for 8 Å) (none >15 Å)	-6.19x10 ²⁰ for 3.6 Å (-1.27x10 ²⁰ for 8 Å) (none >15 Å)
PEDOT ₆ ⁺² on Te	-0.186	-1.56x10 ²¹
Neutral PEDOT ₆ on Cu _{1.75} Te	-0.144	-1.14x10 ²¹
PEDOT ₆ ⁺² on Cu _{1.75} Te	-0.239	-2.05x10 ²¹

*PEDOT monomer volume 1.26x10⁻²² cm³

Figure 2 in the manuscript is revised by enhancing the charge density difference by changing isovalue sensitivity from 0.01 to 0.005. We are able to confirm the observation of a decrease in electron density (blue) in the Te phase and increase of charge density (red) at the interface in the revised version.

Figure 2. DFT calculations reveal electronic effects at the organic-inorganic interface. a) Charge density redistribution within polaronic PEDOT hexamer (EDOT₆)²⁺ on the Te surface, as calculated by the difference in total charge density with NW surface charge density and hexamer charge density as subsets b) Electron transfer from Te surface to PEDOT chains monitored by

increase of charge density (red) at the interface and decrease of charge density (blue) at the Te phase

Verification of physical nature of interaction and templating effect at the interface

Importantly, we confirm that there is no chemical bonding, but only physical interaction and charge transfer: the smallest distances between the optimized PEDOT chains and the inorganic surface are calculated, indicating that the distance between PEDOT backbone atoms and either the Te or Cu_{1.75}Te inorganic surfaces is 3.6-4.0 Å, which is larger than the chemical bonding distance for the PEDOT chains.

The sentence

*This weak charge density difference is analogous to other material systems that exhibit physical adsorption.*²⁰

is revised as

“As for the nature of the bonding between the organic and inorganic components, the weak charge density difference and atomic distances between organic and inorganic constituents, calculated to be between 3.6-4.0 Å, are analogous to other material systems that exhibit physical adsorption.¹ This conclusion is further corroborated by the Density of States (DOS) calculated for (i) PEDOT, (ii) the Te surface, and (iii) the hybrid structure (Figure 2, Figure S6, Figure S7 for PDOS), which depicts a trivial change in DOS distribution between the individual and hybrid structures.”

The Table 1 is added in revised manuscript on page 9.

This result is supported by MD simulations of pristine PEDOT on the surface repeated 4 times for different initial structures (Figure S2a). Combining the result from Table 1, it is clear that the first layer of PEDOT is strongly affected by this templating effect on the inorganic surface, which has been implicated in the enhanced conductivity of the organic phase at the organic-inorganic interface and counteracts the de-doping effect (See, K. C. et. al., *Nano Lett.* 2010, 10 (11), 4664–4667).

These studies were carried out for both the Te and planar $\text{Cu}_{1.75}\text{Te}$ surfaces and reveal similar results for both. In simulations, for a 12.12 Å thick layer of $\text{Cu}_{1.75}\text{Te}$, the first layer of PEDOT chains becomes self-aligned at a distance of 4 Å from the surface. The second layer of PEDOT was observed to form at a distance of 8 Å from the surface. The observed de-doping effect (discussed above) is found to be 5 times weaker for the second layer than the first. Accordingly, the third and further layers are even more weakly aligned and do not have any clear layer structure.

A paragraph is added to page 10

“Combined with the MD results, the DFT calculations strongly suggest that the interaction between the PEDOT and Te/ $\text{Cu}_{1.75}\text{Te}$ surface is purely a templating effect; charge transfer does occur at this interface, but no chemical bonding takes place between the organic and inorganic phases.”

A paragraph is added to page 11 line 7.

“We conclude that alignment of PEDOT molecules at the organic-inorganic interface and charge transfer at the interface both play key roles in the high thermoelectric performance observed in the PEDOT:PSS-Te hybrid system, building upon earlier hypotheses proposing increased electrical conductivity at the interface.²”

4) The authors should include the equation for η in the text and should state that it is actually the reduced chemical potential. The reduced chemical potential will largely depend on doping, but it is not solely determined by the number of charge-carriers as the authors current description implies. The authors need to also state how this reduced chemical potential was determined. Also, it would be helpful if the authors suggest how a small amount of Cu may decrease η .

Response: We thank to Reviewer for their sharp comment. We agree that a distinction should be made between doping and reduced chemical potential, which is a more complex concept. Accordingly, we have restated η as the reduced chemical potential in the revised manuscript. For crystalline degenerately doped semiconductors, the reduced chemical potential will depend upon the carrier concentration and the band effective mass. Here, since σ_{E_0} is constant along a fit line

and represents a term that is related to the band effective mass, η can be considered to vary only with the charge carrier concentration. We appreciate the Reviewer's close read and believe that the manuscript has been clarified and the rigor of our language has been improved by rephrasing the wording in the manuscript accordingly. To re-iterate, we have calculated η value using following Eq.

$$S = \frac{k_B}{e} \left[\frac{(s+1)F_s(\eta)}{sF_{s-1}(\eta)} - \eta \right] \quad (1)$$

Where S is the experimental Seebeck coefficient value for a particular value of s . $F_s(\eta)$ is the Fermi integral. For a detailed explanation of why a decrease in η is observed in the low Cu loading regime, please see the response to the Reviewer's question 1.

5) It is a big stretch to say that the P3HT-CNT data lies on the purple curve shown in Figure 5a. I recommend removing this data series and seeing if a similar set of data in the literature may provide data over a larger S and conductivity range that can be used instead.

Response: We thank the Reviewer for their helpful suggestion. We agree that it is important to include sufficient and appropriate data in this modeling effort. As suggested, we have gone through literature in detail and identified several additional papers detailing P3HT-CNT based hybrid systems. We have revised the figure in question to include additional data from this body of literature, and we believe it demonstrates sufficient breadth to support our conclusions. Both single wall CNT-based hybrids^{35, 37} (open stars) and multiwall CNT-based hybrids^{34, 36} (close triangles), are consistent with our conclusions, demonstrating good $s=3$ fit with same σ_{E_0} values at the higher doping range. Note, for the single walled CNTs there seems to be a reduction in σ_{E_0} at lower doping ranges (10^{-4} - 10^{-1} S/cm) but the $s=3$ trend remains consistent.

Typically, the Seebeck coefficients for samples in a low conductivity data range (10^{-5} to 10^{-3} S/cm) have large error bars, and therefore the reliability of such data may be low. On the other hand, the work done by C. Bounioux et. al.,³ on SWCNTs-P3HT (green horizontal triangles), quite interestingly, does seem to deviate from the $s=3$ curve. Strong π - π interactions have been reported

between CNTs and P3HT.³⁹ It is possible that these stronger chemical interactions between CNTs and P3HT result in deviations for the expected S - σ dependence. While this is interesting and warrants further exploration, we believe that, combined, the many literature data provide sufficient basis for our conclusions. We would like to emphasize that the results on P3HT-Bi₂Te₃ (He, M. *et. al.*, *Energy Environ. Sci.* **5**, 8351–8358 (2012)) are consistent with the results for the PEDOT:PSS-CuTe system and corroborate our conclusions. We appreciate the Reviewer’s helpful suggestion, and believe the quality of the manuscript has been improved based on this additional work. Accordingly, we have added the following sentences to page 20, line 8, of the revised manuscript:

“In the case of P3HT:SWCNT(MWCNT) hybrid systems, while the sparse literature data available lies on the $s=3$ curve with different value of σ_{E_0} , some do indeed deviate for higher values of conductivity. There is a possibility of strong π - π interactions between CNTs and P3HT, which has been hypothesized to cause deviation from the $s=3$ curve³⁹ ”

Figure 5 (a) The electrical conductivity vs Seebeck coefficient data of F4TCNQ doped P3HT (closed circles)³, Fe((CF₃SO₂)₂N)₃ doped P3HT (open squares)⁴, highly aligned P3HT with trichlorobenzene (closed pentagons)⁵, P3HT:MWCNT (closed triangles)³⁶, P3HT:SWCNT (open star)^{35, 37} and P3HT:Bi₂Te₃ (closed star)³⁸ hybrid systems. It is seen that experimental data of pure P3HT and P3HT:Bi₂Te₃ lies on $s=3$ curve, consistently identical for the hybrid and the pure polymeric systems. For P3HT-SWCNT(MWCNT)

based hybrid system, while some literature data lies on $s=3$ curve, a deviation is observed for few literature data.

6) More details on the measurement geometry for the Seebeck coefficient measurements need to be given. Improper measurement geometries can lead to large errors in the Seebeck coefficient, see van Reenen and Kemerink, *Organic Electronics*, 2014, 15, 2250. More details on the temperature dependent electrical conductivity and Seebeck measurements should also be provided – specifically electrode geometries and distances.

Response- We thank the Reviewer for this suggestion. We agree that proper measurement methodologies are critical for obtaining reliable and reproducible results. The details of our measurement technique have been published recently in Pawan Kumar et. al., *Review of Scientific Instruments* 88, 125112 (2017). We are confident in the techniques used in this work, especially given that the details of the technique have already undergone peer review. We have taken care of all the errors mentioned in literature “van Reenen and Kemerink, *Organic Electronics*, 2014, 15, 2250”.

Additional points:

1.) In the caption of Figure 4, the authors state that the doping level (again, doping level should not be used) only changes <30%. This statement is misleading since the value appears to reach a maximum of 1.4 and a minimum of 0.7. Statement needs to be reworded and/or the value changed.

Response- We thank the Reviewer for their comment. We have changed the wording from “doping” to “reduced chemical potential”. The 30% change in reduced chemical potential was estimated relative to the room temperature value (it changes from 1 to 1.2-1.4 for all samples). We have rephrased the wording in the revised manuscript to reflect this with more clarity on page 18, line 9:

“The reduced chemical potential with respect to room temperature value, η/η_{300K} , does not change significantly with lowering temperature as shown in Figure 4b (24%, 35% and 40% for 0, 10 and 50 % Cu loading respectively from room temperature value).”

2.) It seems that two different PEDOT-Te interaction energies are reported, 160 kcal/mol on line 148 and 88 kcal/mol on line 159-160. These should either be corrected or the reasons for these differences clarified.

Response: We thank the Reviewer for pointing out an opportunity to enhance the clarity of the manuscript. These two reported interaction energies are indeed different figures. We agree that the difference between the two calculations should be clear to the reader, and we have revised the manuscript accordingly.

These two values represent interaction energies between PEDOT and Te in different environments. In the first, we are investigating the interaction energy between PEDOT and Te in the presence of PSS; the cell used in this simulation contains both PEDOT and PSS moieties in different configurations. This environment is depicted in Figure S4.

Figure S4: Comparison of interaction energies for two morphologies for a) six PEDOT and b) three PSS oligomer chains on the Te nanowire interface.

In the second calculation, we investigate the interaction of PEDOT and Te without any PSS present (pristine PEDOT environment). This environment was shown in Figure S5 and resulted in an interaction energy of 88 kcal/mol.

Figure S5: One of the equilibrium structure from MD simulations demonstrated a) self-assembly and self-alignment of PEDOT chains on Te surface, b) self-alignment of PEDOT chains on Cu_{1.75}Te (no self-assembly) surface.

In addition to rephrasing the discussion regarding these two calculations, we also performed more MD simulations for the first scenario, to improve the quality of the results in our work, and revised Figure S4 accordingly. In the original manuscript, simulation cells only contained a single PSS chain, and the result was that the PEDOT species would move to the inorganic surface and displace the PSS chain. To remove this effect and more accurately calculate the interaction energy between PEDOT moieties and the inorganic surface when PSS is present in between, simulations were performed including three chains of PSS₂₄ and six chains of PEDOT₁₂. This simulation and result are incorporated in the revised Figure S4. In addition, number of atom layers in MD simulations increased from six to ten for Te NW and four to eight for Cu_{1.75}Te NW to model more realistic experimental conditions at the interface (Figure1, Figure S1-S4).

The 3rd paragraph on page 7 is revised as:

“To investigate the driving force for PEDOT alignment (and lack thereof for PSS) on the inorganic surface, the interfacial PEDOT-inorganic and PSS-inorganic interaction energies were calculated and compared. Two cells were equilibrated with six π -stacked PEDOT and three PSS oligomers (details in Methods, Fig. S4). In one of the cells, PEDOT chains are placed at the organic-inorganic interface; in the other, PSS chains. The polymer-Te interaction energy is determined to be -423 kcal/mol for PEDOT layers on the Te NW surface and -191 kcal/mol for the PSS oligomer on the Te NW surface. This result indicates a thermodynamic driving force for self-assembly of PEDOT over PSS on the nanowire surface, consistent with the structures observed in the MD simulations described previously. The interaction of the same systems with a $\text{Cu}_{1.75}\text{Te}$ NW is about two times stronger calculated as -792 and -388 kcal/mol for PEDOT layers on the NW surface and PSS layers on the NW surface respectively.”

The sentence in parenthesis at line 159-160 in the original manuscript: *“(131 kcal/mol interaction between PEDOT chains and $\text{Cu}_{1.75}\text{Te}$ vs. 88 kcal/mol interaction between PEDOT chains and Te)”* has been deleted to avoid any further confusion for different PEDOT interaction energies with surface.

The sentence at page 7 line 31 is revised as

“We attribute this phenomenon to stronger interaction between PEDOT and the $\text{Cu}_{1.75}\text{Te}$ surface, resulting in reduced movement of the PEDOT chains on the $\text{Cu}_{1.75}\text{Te}$ surface (Video S6, S7).”

3.) At one point the authors state that they are below the percolation threshold in the PEDOT:PSS-Te(Cu) hybrids, but from the SEM images this does not appear to be the case. It would be helpful if the authors could state both the volume and weight % of the NWs in the films, as well as the percolation threshold.

Response - We thank the Reviewer for their close read of the paper. In the previous version of the paper, there was an error stating that the hybrid films reported were below the percolation threshold. We have corrected this in the revised manuscript. In fact, these films are above the classical percolation threshold. This can be seen in the SEM images provided, as the Reviewer correctly pointed out. The loading of nanowires in the films is typically 60-70 wt%, corresponding

to 30-42 vol%. However, this does not affect the discussion or conclusions we put forth in the paper. Despite the high inorganic loading in these films, the electrical and thermoelectric properties are dominated by the organic component, as discussed in more detail in response to the Reviewer's second question.

4.) The authors should further explain where the various lines presented in Figure 3b originate from (degenerate, non-degenerate, and CT). Presumably CT stands for charge transport model and the authors are using this to denote that this is the complete model, but this should be listed.

Response- We thank the Reviewer for their thoughtful comment. The degenerate and non-degenerate lines originate from the Kang-Snyder charge transport model itself by applying degenerate ($\eta \gg 1$) and non-degenerate ($\eta \ll 1$) conditions. From Kang-Snyder model, conductivity and Seebeck can be written as:

$$\sigma = \sigma_{E_0}(T) \times sF_{s-1}(\eta) \quad (1)$$

$$S = \frac{k_B}{e} \left[\frac{(s+1)F_s(\eta)}{sF_{s-1}(\eta)} - \eta \right] \quad (2)$$

In degenerate region when ($\eta \gg 1$), conductivity and Seebeck can be written as :

$$\sigma = \sigma_{E_0}(T) \times \eta^s \quad (3)$$

$$S = \frac{k_B}{e} \frac{\pi^2}{3} s \left(\frac{\sigma}{\sigma_{E_0}} \right)^{-\frac{1}{s}} \quad (4)$$

In non-degenerate region when ($\eta \ll 1$), conductivity and Seebeck can be written as:

$$\sigma = \sigma_{E_0}(T) \times sT(s) \exp(\eta) \quad (5)$$

$$S = \frac{k_B}{e} \left[s + 1 - \ln\left(\frac{\sigma}{\sigma_{E_0}sT(s)}\right) \right] \quad (6)$$

We have added this part in the SI.

5.) The sentence “organic semiconductors are successfully modeled as semiconductors” is odd – they are semiconductors and of course can be modelled as such. Is this meaning to say that they are modeled using theories originally developed for inorganic semiconductors?

Response: We thank to Reviewer for pointing out this instance of confusing language. We have edited the following sentence in page 3 line 28 to make our point more clear.

"In this model, the thermoelectric transport of conducting polymers has been modeled using energy independent parameter, σ_{E0} and energy dependent parameter 's' over a large range of conductivity."

Overall, we thank the reviewer for their deep insights and we believe that our manuscript is much improved after addressing the questions.

Reviewer #2 (Remarks to the Author):

This paper presents detailed theoretical and experimental results on thermoelectric properties of PEDOT:PSS-Te/CuTe nanowire. It concludes that there is no modification of the energy dependence of scattering or charge transfer between polymer and the inorganic nanowire. The Kang-Snyder equation was used to explain the transport properties of the hybrid material, based on the assumption that the transport properties are dominated by the polymers. The work is carefully done and the conclusions would be informative for the researchers in the field of hybrid organic-inorganic thermoelectric materials.

1. (a) It is already known that Tellurium has very high thermoelectric performance,[Nat. Commun. 7, 10287 (2016)], so does it contribute to the thermoelectric performance of the hybrid material? Cu_xTe also has very high power factor (reference [25]). How to exclude its effect from the total thermoelectric performance of the hybrid material.

Response- We thank the Reviewer for their insightful comment. It is clear that this is a key point to clarify for our revision, especially given that a similar question has been raised by Reviewer 1 in Q2. We believe that our response to Reviewer 1's Q2 addresses this point as well, and have included it again below.

“The $s=1$ fit is a classic indicator for transport in PEDOT relative to other semiconducting polymers, and for consistency with the organic thermoelectrics literature, this point is stressed. However, this is not sufficient to prove that transport occurs predominantly through the PEDOT phase in organic-inorganic composite films. Thus, we have performed additional experimentation, which, when combined with the data presented in our paper, provides further support for this claim. We discuss this conclusion in response to question number 3 below, with a focus on charge transfer between the inorganic and organic phases.

In addition to the PEDOT:PSS-Te/CuTe NWs, we synthesized Te/CuTe NWs in an insulating polymer matrix (polyvinylpyrrolidone (PVP), 55 kDa), keeping the polymer-to-inorganic ratio in each film roughly the same by mass as for the PEDOT composites. We performed electrical and

thermoelectric transport measurements, and modeling according to the Kang-Snyder model (Fig S17 below). The results of this experiment shows that the transport properties in these composite films cannot be explained as the result of transport in a percolated network of inorganic nanostructures, instead suggesting a key role for the organic component.

Fig. S17. Experimental data of Seebeck (S) vs conductivity (σ) for Te/CuTe nanowire embedded in insulating polymer matrix (closed square), PEDOT:PSS-Te(Cu_x) NW hybrid system (closed circles) and bulk Te (closed star) modelled with $s=1$ (solid lines). It is seen Seebeck vs conductivity data of PEDOT-based hybrid system and bulk Te lies on $s=1$ curve with different σ_{E_0} transport coefficient values. In case of Te/CuTe nanowire embedded in insulating polymer matrix, the Seebeck vs conductivity data lies on all possible s values ($s=1, 2,$ and 3) and it is difficult to distinguish that which ‘ s ’ is true for the system.

If conductivity in these hybrid systems were described by transport through a percolated inorganic network, then one would expect the measured conductivity to be relatively independent of the organic matrix. However, for the PVP-Te NW system, the electrical conductivity is drastically reduced relative to the PEDOT-Te NW system or even to bulk Te; the conductivity of PVP-Te NWs is 0.015 S/cm, which is approximately 1000 times lower than that of bulk Te. Even at maximum Cu loading, the maximum conductivity measured for PVP-Te/CuTe NWs is 0.7 S/cm, which is also 1000 times lower than bulk $Cu_{1.75}Te$ (B. A. Mansour et. al., *Thin Solid Films*, 247, 112 (1994)) and $Cu_{1.75}Te$ nanowires (C. Zhou et. al., *ACS Appl. Mater. Interfaces*, 2015, 7, 21015

(2015)). This low conductivity in the PVP-Te/CuTe supports our conclusion that the organic component plays a key role in transport in these hybrid films. To investigate this phenomenon further, we have plotted the transport properties for PVP-Te/CuTe NW, PEDOT-Te/CuTe NW, and bulk Te on Fig S17 below, and fit these data using the Kang-Snyder model. Upon applying the model, we found that, while bulk Te shows $s=1$ dependence with large values of σ_{E_0} , it is impossible to distinguish which s value is true for our Te/CuTe NW in insulating polymer matrix, unlike the clear demarcation in the Te/CuTe-PEDOT system. It is well known that scattering of charges is dominated by acoustic phonon scattering at room temperature for inorganic semiconductors which corresponds to $s=1$ ($r=-0.5$).² Therefore, $s=1$ dependence is expected in bulk Te (green stars). Hence, if the charge transport was dominated through a percolation pathway through the semiconducting nanostructures, we should expect $s=1$ for all inorganic-organic hybrid systems. However, as described in the original manuscript, this is not the case for the majority of non-PEDOT based hybrid systems. For example, in P3HT based hybrid systems (Fig. 5(a) in the revised manuscript), $s=3$ dependence is conclusively observed, confirming that transport in the polymer matrix is indeed the key factor in the overall film thermoelectric properties.”

(b) If there is no charge transfer between the polymer and the Te/Cu_{1.75}Te nanowire, why the Seebeck coefficients of PEDOT:PSS-TE and PEDOT:PSS-Cu_{1.75}Te show a big difference (line238).

Response- We are thankful for the insightful perspective of the Reviewer, and note again that this concern is tied closely to Reviewer 1’s Q3. We have included our response below, which we believe addresses this issue and have revised the manuscript accordingly.

We have performed additional calculations to extensively consider charge redistribution at the inorganic-organic interface. The results provide further insight into the physics of carrier transport in the system, and corroborate our conclusions that the thermoelectric behavior in these hybrid systems is dominated by charge transport in the organic phase. We also supplement the evidence provided for the physical nature of the interaction between the PEDOT chains and the inorganic surface.

² D. A. Neamen, “Semiconductor Physics And Devices” 3rd edition, Macgraw Hill (2003).

Using a Mulliken charge analysis, charge transfer rates were calculated for PEDOT hexamers on both Te and $\text{Cu}_{1.75}\text{Te}$ surfaces. Moreover, pristine state (neutral PEDOT) and bipolaronic/charged PEDOT conditions were considered as the surface PEDOT structures. For the bipolaron case, a +1 charge is added for every three monomers, as predicted experimentally (*G. Zotti, S. Zecchin, and G. Schiavon, F. Louwet and, L. Groenendaal, X. Crispin, W. Osikowicz, W. Salaneck and M. Fahlman, Macromolecules, 2003, 36, 3337–3344.*). Charge transfer rates are then calculated per monomer (see Table 1 and Table S1). Note that a negative quantity here represents electron transfer from the inorganic surface to the organic PEDOT chains (i.e. hole transfer from the organic PEDOT chain to the inorganic surface). As a result of these calculations, it is evident that excited state charge transfer does indeed take place between the organic and inorganic phases. In every case, this charge transfer provides a strong de-doping effect of holes in the p-type PEDOT chains. This de-doping plays a key role in understanding the thermoelectric trends in these hybrid systems, as is discussed in depth above in response to the Reviewer's first question.

A new paragraph is added to the revised manuscript at page 8, line 20:

“The maximum charge transfer rates from the inorganic surface to the first layer of (neutral) PEDOT chains on the surface were determined to be -0.078 and -0.144 for Te and $\text{Cu}_{1.75}\text{Te}$, respectively. Charge transfer rates are higher for the charged (EDOT_6)⁺² bipolaron, calculated to be -0.186 and -0.239 for Te and $\text{Cu}_{1.75}\text{Te}$, respectively (Table S1). Note that a negative quantity here represents electron transfer from the inorganic surface to the organic PEDOT chains (i.e. hole transfer from the organic PEDOT chain to the inorganic surface). In every case, this charge transfer provides a de-doping effect of holes in the p-type PEDOT chains, which plays a key role in understanding the thermoelectric trends in these hybrid systems (Table 1). This de-doping effect is stronger for the doped PEDOT bipolaron compared to pristine PEDOT chains and also stronger for PEDOT on the $\text{Cu}_{1.75}\text{Te}$ surface compared to PEDOT on the Te surface. The charge transfer and de-doping effect is only observed for the first two layer of PEDOT chains and vanished for the higher distances.”

Table 1. De-doping level of neutral and doped PEDOT hexamer by Te and Cu_{1.75}Te surfaces for the geometry optimized structures.

	De-doping effect* (electron/cm³)
Neutral PEDOT ₆ on Te	-6.19x10 ²⁰ for 3.6 Å (-1.27x10 ²⁰ for 8 Å) (none >15 Å)
PEDOT ₆ ⁺² on Te	-1.56x10 ²¹
Neutral PEDOT ₆ on Cu _{1.75} Te	-1.14x10 ²¹
PEDOT ₆ ⁺² on Cu _{1.75} Te	-2.05x10 ²¹

*PEDOT monomer volume 1.26x10⁻²² cm³

De-doping due to interfacial charge transfer:

A new table has been added to the SI:

Table S1. Charge transfer and de-doping effect for the pristine and doped PEDOT chains on Te and Cu_{1.75}Te surfaces for the geometry optimized structures.

	electron/monomer (Mulliken)	De-doping effect* (electron/cm³)
Neutral PEDOT ₆ on Te	-0.078 for 3.6 Å (-0.016 for 8 Å) (none >15 Å)	-6.19x10 ²⁰ for 3.6 Å (-1.27x10 ²⁰ for 8 Å) (none >15 Å)
PEDOT ₆ ⁺² on Te	-0.186	-1.56x10 ²¹
Neutral PEDOT ₆ on Cu _{1.75} Te	-0.144	-1.14x10 ²¹
PEDOT ₆ ⁺² on Cu _{1.75} Te	-0.239	-2.05x10 ²¹

*PEDOT monomer volume 1.26x10⁻²² cm³

Figure 2 in the manuscript is revised by enhancing the charge density difference by changing isovalue sensitivity from 0.01 to 0.005. We are able to confirm the observation of a decrease in electron density (blue) in the Te phase and increase of charge density (red) at the interface in the revised version.

Figure 2. DFT calculations reveal electronic effects at the organic-inorganic interface. a) Charge density redistribution within polaronic PEDOT hexamer $(\text{EDOT}_6)^{2+}$ on the Te surface, as calculated by the difference in total charge density with NW surface charge density and hexamer charge density as subsets b) Electron transfer from Te surface to PEDOT chains monitored by increase of charge density (red) at the interface and decrease of charge density (blue) at the Te phase

Verification of physical nature of interaction and templating effect at the interface

Importantly, we confirm that there is no chemical bonding, but only physical interaction and charge transfer: the smallest distances between the optimized PEDOT chains and the inorganic surface are calculated, indicating that the distance between PEDOT backbone atoms and either the Te or $\text{Cu}_{1.75}\text{Te}$ inorganic surfaces is 3.6-4.0 Å, which is larger than the chemical bonding distance for the PEDOT chains.

The sentence

*This weak charge density difference is analogous to other material systems that exhibit physical adsorption.*²⁰

is revised as

“As for the nature of the bonding between the organic and inorganic components, the weak charge density difference and atomic distances between organic and inorganic constituents, calculated to be between 3.6-4.0 Å, are analogous to other material systems that exhibit physical adsorption.¹ This conclusion is further corroborated by the Density of States (DOS) calculated for (i) PEDOT, (ii) the Te surface, and (iii) the hybrid structure (Figure 2, Figure S6, Figure S7 for PDOS), which depicts a trivial change in DOS distribution between the individual and hybrid structures.”

The Table 1 is added in revised manuscript in page 9.

This result is supported by MD simulations of pristine PEDOT on the surface repeated 4 times for different initial structures (Figure S2a). Combining the result from Table 1, it is clear that the first layer of PEDOT is strongly affected by this templating effect on the inorganic surface, which has been implicated in the enhanced conductivity of the organic phase at the organic-inorganic interface and counteracts the de-doping effect (See, K. C. et. al., *Nano Lett.* 2010, 10 (11), 4664–4667).

These studies were carried out for both the Te and planar Cu_{1.75}Te surfaces and reveal similar results for both. In simulations, for a 12.12 Å thick layer of Cu_{1.75}Te, the first layer of PEDOT chains becomes self-aligned at a distance of 4 Å from the surface. The second layer of PEDOT was observed to form at a distance of 8 Å from the surface. The observed de-doping effect (discussed above) is found to be 5 times weaker for the second layer than the first. Accordingly, the third and further layers are even more weakly aligned and do not have any clear layer structure.

A paragraph is added to page 10

“Combined with the MD results, the DFT calculations strongly suggest that the interaction between the PEDOT and Te/Cu_{1.75}Te surface is purely a templating effect; charge transfer does occur at this interface, but no chemical bonding takes place between the organic and inorganic phases.”

(c) It is not clear how the loading of copper increases the carrier concentration of the polymer.

Response: We thank the Reviewer for their thoughtful point. This is indeed an important and interesting point. While it is difficult to directly probe the doping mechanism, it is most likely that positively charged Cu ions remain in the PEDOT:PSS phase during the Cu addition, and these ions interact chemically with PEDOT moieties to contribute to the increase in carrier concentration within the organic phase. In our original paper describing the PEDOT-Te/Cu_{1.75}Te system, we report synchrotron Cu L_{2,3} XAS data which provides direct evidence for the presence of both Cu⁺¹ and Cu⁺² species in these films.⁷ While Cu⁺¹ contributes to the formation of Cu_{1.75}Te alloy phase, the Cu⁺² ions likely remain within the organic phase. It has been shown that positive ions can react with thiophene moieties on PEDOT to form polaronic or bipolaronic carriers, and it is likely via this same mechanism that the copper loading increases the PEDOT carrier concentration.⁸

2. Pure organic PEDOT:PSS has already achieved a very high ZT value of 0.25-0.42.[Nat. Mater. 10, pages 429–433 (2011) & Nat. Mater. 12, pages 719–723 (2013)] However, the current work is mainly based on a hybrid PEDOT:PSS-Te composite with ZT=0.1. The multiphase approach does not show advantage over the pure organic materials on the thermoelectric performance.

Response: We thank the Reviewer for their comment. The current manuscript is not narrowly focused on optimizing ZT for a particular system, but rather further elaborating a microscopic model of hybrid systems, informed by experimental data and a theoretical framework that broadly applies to this very large (and growing) class of thermoelectrics. The motivation of this work is to use a combination of detailed simulations, careful theoretical modeling with comprehensive experimental measurements to understand the physics of the interface between inorganic-organic hybrid systems and its effect on thermoelectric charge transfer. Indeed, the insights from this work will spur researchers further to create new multiphase materials that will be superior to the organic and inorganic constituents, but this is beyond the scope of our current work. Moreover, those ZT values approaching 0.42 for PEDOT:PSS have not been broadly reproduced and seem to be highly sensitive to experimental conditions.

3. The paper provides a deep understanding on the transport properties of PSS-PEDOT/Te nanowire that has been previously reported. It is quite strange that Figure 3a in this paper is very close to the Figure 3a in the previous work of the authors (Reference [10]). Future directions for

hybrid organic-inorganic thermoelectric materials have been pointed out, but it seems sufficiently striking insights are lacking.

Response- We appreciate the Reviewer's acknowledgement that we provide a deep understanding of the transport properties that have been previously reported. Our current manuscript represents a collaboration including Zaia et al., the original authors of Reference [10]. Because of this, we used the same room temperature data for our study which are reported in ref. 10. Appropriate permissions have been obtained and the caption has been edited to reflect this in the revised manuscript. In addition to this data, we have performed careful temperature dependent experiments that are new to this manuscript and complement our theoretical understanding. We believe that a detailed understanding of the state-of-the-art in hybrid thermoelectric materials, as we've provided in this treatise, is a significant step towards creating next-generation hybrid materials. In the revised conclusion, we have laid out our key insights, which are augmented by the two Reviewers' detailed comments. We now show, unequivocally, that the thermoelectric transport can be explained by a combination of templating of the polymer on the nanowire surface, de-doping due to charge transfer at the interface, and finally doping due to Cu⁺ loading. Thus, any future improvements for hybrid materials (similar to the P3HT-CNT hybrid) will come from chemical interactions (not physical, as is the case in our work) for *s*=3 polymers, pushing the envelope in this field. This insight is vital to further improvements in this burgeoning field.

References

1. Jarvis, S. P. *et al.* Physisorption Controls the Conformation and Density of States of an Adsorbed Porphyrin. *J. Phys. Chem. C* **119**, 27982–27994 (2015).
2. See, K. C. *et al.* Water-Processable Polymer–Nanocrystal Hybrids for Thermoelectrics. *Nano Lett.* **10**, 4664–4667 (2010).
3. Glauddell, A. M., Cochran, J. E., Patel, S. N. & Chabinyk, M. L. Impact of the Doping Method on Conductivity and Thermopower in Semiconducting Polythiophenes. *Adv. Energy Mater.* **5**, 1401072 (2015).

4. Zhang, Q., Sun, Y., Xu, W. & Zhu, D. What To Expect from Conducting Polymers on the Playground of Thermoelectricity: Lessons Learned from Four High-Mobility Polymeric Semiconductors. *Macromolecules* **47**, 609–615 (2014).
5. Qu, S. *et al.* Highly anisotropic P3HT films with enhanced thermoelectric performance via organic small molecule epitaxy. *NPG Asia Mater.* **8**, e292 (2016).
6. Taek Hong, C., Hun Kang, Y., Ryu, J., Yun Cho, S. & Jang, K.-S. Spray-printed CNT/P3HT organic thermoelectric films and power generators. *J. Mater. Chem. A* **3**, 21428–21433 (2015).
7. Zaia, E. W. *et al.* Carrier Scattering at Alloy Nanointerfaces Enhances Power Factor in PEDOT:PSS Hybrid Thermoelectrics. *Nano Lett.* **16**, 3352–3359 (2016).
8. Park, B. *et al.* Neutral, Polaron, and Bipolaron States in PEDOT Prepared by Photoelectrochemical Polymerization and the Effect on Charge Generation Mechanism in the Solid-State Dye-Sensitized Solar Cell. *J. Phys. Chem. C* **117**, 22484–22491 (2013).

REVIEWERS' COMMENTS:

Reviewer #1 (Remarks to the Author):

The authors have appropriately addressed most of my concerns and I think that the paper is now much stronger. Composite materials are difficult to model given the large number of variables, but this paper sheds important insight on variables that are not commonly considered when interpreting the thermoelectric performance of conjugated polymer-nanowire blends. The primary conclusion that energy filtering is not likely the culprit for the enhanced Seebeck coefficient and power factor in polymer-nanowire blends is indeed important and should be of interest to those working on composite thermoelectrics.

The addition of Table 2 is helpful in summarizing what is occurring in the different systems.

In Figure S4, I presume that the red molecules stacks of the PSS trimers? If so, they adopt an interesting stacking pattern and it may be worth pointing this out in the SI below the figure.

Reviewer #2 (Remarks to the Author):

I have no further questions based on the response and revisions provided by the authors.

Point-by-point response to Reviewers' questions

Reviewer #1 (Remarks to the Author):

The authors have appropriately addressed most of my concerns and I think that the paper is now much stronger. Composite materials are difficult to model given the large number of variables, but this paper sheds important insight on variables that are not commonly considered when interpreting the thermoelectric performance of conjugated polymer-nanowire blends. The primary conclusion that energy filtering is not likely the culprit for the enhanced Seebeck coefficient and power factor in polymer-nanowire blends is indeed important and should be of interest to those working on composite thermoelectrics.

We thank the Reviewer for the valuable comments and support of our work. Indeed, they have captured the essence of our paper well.

The addition of Table 2 is helpful in summarizing what is occurring in the different systems.

We are glad that this change clarifies the detailed physics in the hybrid system and is helpful.

In Figure S4, I presume that the red molecules stacks of the PSS trimers? If so, they adopt an interesting stacking pattern and it may be worth pointing this out in the SI below the figure.

We thank the Reviewer for their comment – indeed this figure does not represent a stacking pattern or an ordering of the PSS, instead it is just an illustration of the PSS surrounding the PEDOT layer. We've made a change in the figure caption that reflects this:

"PEDOT and PSS chains are represented by blue and red in color, respectively" to the first paragraph in Supplementary Notes.

Further, we added the discussion as below:

"Interaction energies calculated for six EDOT12 and three SS24 oligomers with two different configurations where surface is PEDOT rich and PSS rich, respectively. High difference in the interaction energies showed that concentration of PEDOT chains are higher at the interface compared to PEDOT:PSS bulk which has higher PSS concentration. Taking into consideration with results from Figure 1 and Figure S2, nanowire-PEDOT:PSS interface is not only PEDOT rich, but also highly self organized and self aligned PEDOT chains present at the interface which is one the origin of the enhanced TE performance of PEDOT:PSS hybrid materials."

Reviewer #2 (Remarks to the Author):

I have no further questions based on the response and revisions provided by the authors.

We thank the Reviewer for their support of our work.